# Clinical Evidence for Thermometric Parameters to Guide Hyperthermia Treatment

**DOI:** 10.3390/cancers14030625

**Published:** 2022-01-26

**Authors:** Adela Ademaj, Danai P. Veltsista, Pirus Ghadjar, Dietmar Marder, Eva Oberacker, Oliver J. Ott, Peter Wust, Emsad Puric, Roger A. Hälg, Susanne Rogers, Stephan Bodis, Rainer Fietkau, Hans Crezee, Oliver Riesterer

**Affiliations:** 1Center for Radiation Oncology KSA-KSB, Cantonal Hospital Aarau, 5001 Aarau, Switzerland; adela.ademaj@ksa.ch (A.A.); dietmar.marder@ksa.ch (D.M.); emsad.puric@ksa.ch (E.P.); roger.haelg@ksa.ch (R.A.H.); susanne.rogers@ksa.ch (S.R.); s.bodis@bluewin.ch (S.B.); 2Doctoral Clinical Science Program, Medical Faculty, University of Zurich, 8032 Zürich, Switzerland; 3Department Radiation Oncology, Charité-Universitätsmedizin Berlin, Corporate Member of Freie Universität Berlin and Humboldt-Universität zu Berlin, 13353 Berlin, Germany; paraskevi-danai.veltsista@charite.de (D.P.V.); pirus.ghadjar@charite.de (P.G.); eva.oberacker@charite.de (E.O.); peter.wust@charite.de (P.W.); 4Department of Radiation Oncology, Universitätsklinikum Erlangen, 91054 Erlangen, Germany; oliver.ott@uk-erlangen.de (O.J.O.); rainer.fietkau@uk-erlangen.de (R.F.); 5Comprehensive Cancer Center Erlangen-EMN, 91054 Erlangen, Germany; 6Institute of Physics, Science Faculty, University of Zurich, 8057 Zurich, Switzerland; 7Department of Radiation Oncology, University Hospital Zurich, University of Zurich, 8091 Zurich, Switzerland; 8Department of Radiation Oncology, Amsterdam UMC, University of Amsterdam, Cancer Center Amsterdam, 1105 AZ Amsterdam, The Netherlands; h.crezee@amsterdamumc.nl

**Keywords:** hyperthermia, thermometric parameters, preclinical data, clinical evidence

## Abstract

**Simple Summary:**

Hyperthermia (HT) is a promising therapeutic option for multiple cancer entities as it has the potential to increase the cytotoxicity of radiotherapy (RT) and chemotherapy (CT). Thermometric parameters of HT are considered to have potential as predictive factors of treatment response. So far, only limited data about the prognostic and predictive role of thermometric parameters are available. In this review, we investigate the existing clinical evidence regarding the correlation of thermometric parameters and cancer response in clinical studies in which patients were treated with HT in combination with RT and/or CT. Some studies show that thermometric parameters correlate with treatment response, indicating their potential significance for treatment guidance. Thus, the establishment of specific thermometric parameters might pave the way towards a better standardization of HT treatment protocols.

**Abstract:**

Hyperthermia (HT) is a cancer treatment modality which targets malignant tissues by heating to 40–43 °C. In addition to its direct antitumor effects, HT potently sensitizes the tumor to radiotherapy (RT) and chemotherapy (CT), thereby enabling complete eradication of some tumor entities as shown in randomized clinical trials. Despite the proven efficacy of HT in combination with classic cancer treatments, there are limited international standards for the delivery of HT in the clinical setting. Consequently, there is a large variability in reported data on thermometric parameters, including the temperature obtained from multiple reference points, heating duration, thermal dose, time interval, and sequence between HT and other treatment modalities. Evidence from some clinical trials indicates that thermal dose, which correlates with heating time and temperature achieved, could be used as a predictive marker for treatment efficacy in future studies. Similarly, other thermometric parameters when chosen optimally are associated with increased antitumor efficacy. This review summarizes the existing clinical evidence for the prognostic and predictive role of the most important thermometric parameters to guide the combined treatment of RT and CT with HT. In conclusion, we call for the standardization of thermometric parameters and stress the importance for their validation in future prospective clinical studies.

## 1. Introduction

Hyperthermia (HT) is a clinical treatment for cancer which extraneously and intrinsically heats malignant cells to a temperature of 40–43 °C for a suitable period of time [1,2]. Heat delivered to tumor tissues can act as a cytotoxic or sensitizing agent to enhance their remission or at least regression by utilizing several biological mechanisms and pleiotropic effects when combined with other conventional cancer treatment techniques, such as radiotherapy (RT) and/or chemotherapy (CT).

The biological effects of HT, which all favor its use in combination with RT and CT, include direct cytotoxicity, radiosensitization, chemosensitization, and immune modulation. HT-induced cell lethality is predominantly a result of conformational changes and the destabilization of macromolecule structures including the disruptions in cell metabolism, inhibition of DNA repair, and triggering of cellular apoptotic pathways [3,4,5,6]. The direct HT-induced cell lethality is known to be intrinsically tumor-selective for hypoxic cells [7]. During heating, enhanced blood perfusion in tumor tissues influences the radiosensitizing and chemosensitizing effects of HT by increasing the tumor oxygenation level and local concentration of CT drugs respectively [4,8,9]. Radiosensitization and chemosensitization effects, as well as the inhibition of DNA synthesis and repair, on the molecular level depend on the aggregation of proteins produced by HT-induced denaturation [10]. Moreover, protein unfolding and the intracellular accumulation of proteins trigger molecular chaperones including the heat shock proteins (HSPs) [11]. The release of HSPs and other “immune activating signals” underly the inflammatory and immunogenic responses to HT in combination with RT and/or CT and can promote anti-tumor immunity [12,13,14]. Exploiting molecular and physiological mechanisms evoked by HT can improve the efficacy of RT and CT. Therefore, HT in cancer treatment is used mainly within the framework of multimodal treatment strategies [3,8].

Multiple preclinical studies have been designed to unravel the relationship between biological mechanisms induced by HT and thermometric parameters as predictors of tumor response [15,16,17,18,19,20]. The parameters investigated in these studies include the temperature achieved during HT [6,15], heating duration, thermal dose [21], time interval between HT and the other treatment modality [15,22,23], the number of HT sessions [24], and the sequence of treatment modality [15,25,26]. All of these parameters were shown to influence the extent to which HT enhances the effect of RT or CT using cellular assays and in vivo models. In addition to thermometric parameters, the treatment parameters of RT and CT, such as total radiation dose, number of RT fractions, type of chemotherapeutic drug and the number of CT cycles, prescribed for a specific clinical indication, also play a significant role in attaining a therapeutic window with synergistic effects when combined with HT [25,27,28].

The effectiveness of HT combined with RT and/or CT has been investigated in many clinical studies with different tumor types. Unfortunately, to date, there is no consensus on HT delivery when combined with these cancer treatment modalities, resulting in substantial heterogeneity of the HT treatment protocols applied. Any comparison of these studies in terms of outcome should be made with caution in view of this heterogeneity in HT protocols. A good understanding of thermometric parameters and their interpretation is mandatory in this regard. However, there is inconclusive clinical evidence about the relationship of thermometric parameters with both tumor and normal tissue responses to HT in combination with RT and/or CT. The reason for this is that thermometric parameters are inconsistently reported or analyzed in prospective clinical studies and the retrospective analyses are conflicting. For instance, minimum tumor temperature was identified as a prognostic factor in a few studies [29,30,31]. However, another study showed that different metrics such as temperature achieved in 90% (T_90_), 50% (T_50_), and 10% (T_10_) in the target volume were more strongly correlated with cancer response than minimum achieved temperature [32]. Furthermore, a short time interval between HT and RT was shown to significantly predict treatment outcome in retrospective analyses of cervical cancer patients [22]. However, conflicting results have been also reported [33] which may be attributed to differences in time interval and tumor temperature achieved, and in patient population [34]. Thermal dose has been successfully tested in several clinical trials as a predictor of tumor response to combined RT and HT treatment [35,36,37,38,39,40,41,42]. These did not result in established thresholds for thermal dose for treating different cancer sites, even though European Society for Hyperthermic Oncology (ESHO) guidelines recommend superficial HT maintains T_50_ ≥ 41 °C and T_90_ ≥ 40 °C [43]. The concept of a relationship between thermometric parameters with treatment outcome is highly attractive because it could improve the understanding of tumor-specific mechanisms of interaction between HT and RT and/or CT. Defining thermometric parameters is therefore important for a meaningful clinical evaluation of HT treatment outcomes when combined with RT and/or CT.

A limited amount of clinical information is available about the effect of thermometric parameters on treatment response. Increasing awareness of the importance of such parameters on the efficacy of HT combined with other cancer treatments is important, and thus these parameters should be evaluated and reported routinely. Achieving the defined thermometric parameters during HT treatment would further increase the effectiveness of biological mechanisms when combined with RT and/or CT. Future prospective clinical studies should include description of all relevant thermometric parameters to pave the way towards the proper analysis and standardization of thermometric parameters for each clinical indication treated with HT in combination with RT and/or CT. 

This work summarizes the evidence underlying thermometric parameters as predictors of treatment outcomes as reported in clinical studies using HT in combination with RT and/or CT for treating different cancer types and emphasizes the need for reference thermometric parameters to improve HT efficacy. For completeness, the findings pertaining to thermometric parameters from preclinical studies are also discussed, to provide comprehensive information about their significance and underlying mechanisms.

## 2. Materials and Methods

### 2.1. Data Sources and Search Strategies

The literature search included databases of clinicaltrials.gov and pubmed.ncbi.nlm.nih.gov from March to September 2021 and randomized prospective and retrospective clinical studies with specific criteria were identified. The search terms were hyperthermia, cancer treatment, randomized clinical studies, prospective clinical studies, and retrospective clinical studies. Those terms were used mainly to search for the title and abstract. We also found articles which were recommended, suggested, or sent to us on the internet. Additionally, we handsearched the reference lists of the most relevant clinical studies and review articles. 

### 2.2. Inclusion and Exclusion Criteria of Clinical Studies 

This non-systematic review included randomized, prospective, and retrospective clinical studies that recruited patients with cancer who were treated with HT and RT and/or CT. The data from randomized trials are only from the patient group which received HT in combination with either RT and/or CT. Data from the non-HT arm were not extracted. 

The main inclusion criteria was the use of either electromagnetic, radiative, or capacitive HT systems, independent of cancer type. Another criterion was more than 10 patients recruited in prospective and retrospective studies. Retrospective studies were only included if analysis of thermometric parameters for HT in combination with RT had been performed. 

Clinical studies which used the thermal ablation technique, interstitial-based/modulated electro HT techniques, interstitial RT techniques, high intensity focused ultrasound (HIFU) HT, whole body HT, and studies in pediatric patients were not included in this review. Pilot and feasibility studies were also excluded. 

### 2.3. Data Extraction and List of Variables Included

The data extracted from the clinical studies contained the following information:First author of the studyStudy design: prospective or retrospectiveRT treatment data: total dose, number of fractionationsCT treatment data: drug and concentration prescribed, number of cyclesThermometric parametersReported clinical endpointsReported relationship between thermometric parameters and clinical endpoint

### 2.4. A Summary of HT Techniques

The clinical studies included in this review administered HT using externally applied power with electromagnetic–based techniques, such as radiofrequency, microwave, or infrared. These techniques differ with regard to their application to treat superficial or deep-seated tumors, as summarized elsewhere [44].

For superficial tumors, the electromagnetic radiative and capacitive systems are the those used in the clinical trials included in this review. The superficial HT techniques and their application are explained in detail elsewhere [43]. The radiative and capacitive systems differ in the way they are applied in the clinic. A study showed that for superficial cancers, the radiative HT system performs better than capacitive systems in terms of temperature distribution [45]. The commercially available radiative superficial systems are the BSD-500 device (Pyrexar Medical, Salt Lake, UT, USA), the ALBA ON4000 (Alba Hyperthermia, Rome, Italy) and contact flexible microwave applicators (SRPC Istok, Fryazino, Moscow region, Russia). Thermotron RF8 (Yamamoto Vinita Co, Osaka, Japan), Oncotherm (Oncotherm Kft., Budapest, Hungary) and Celsius TCS (Celsius42 GmbH, Cologne, Germany) are examples of commercial capacitive systems used for superficial tumors. 

Different HT techniques with unique specifications, characteristics, and limitations are used to treat deep-seated tumors [46]. The ESHO guidelines provide information as to how and when a specific particular HT device should be used to treat deep-seated tumors [46,47]. The radiative HT systems for deep-seated tumors used in clinical trials are the BSD-2000 device (Pyrexar Medical, Salt Lake, UT, USA), the ALBA 4D (Alba Hyperthermia, Rome, Italy), and the Synergo RITE (Medical Enterprises Europe B.V., Amstelveen, The Netherlands), and capacitive systems are Oncotherm (Oncotherm Kft., Budapest, Hungary), Celsius TCS (Celsius 42 GmbH, Cologne, Germany), and Thermotron RF8 (Yamamoto Vinita Co, Osaka, Japan). Another simulation study showed a difference in heating patterns between radiative and capacitive HT for deep-seated tumors [48]. The radiative technique yields more favorable simulated temperature distributions for deep-seated tumors than the capacitive technique.

### 2.5. Definition of Thermometric Parameters 

In this work, the thermometric parameters were extracted from the selected perspective and retrospective clinical studies. The definitions of these parameters are listed in Table 1.

Temperature measurements in the target volume or surrounding tissue are crucial for assessing treatment quality and are represented by temperature metrics. During a HT session, the temperature is usually monitored and recorded using high resistance thermistor probes, fiber optic temperature probes or thermocouples by invasively placing the probes in the target volume or in the vicinity of the target volume [43,46,50]. The ESHO guidelines recommend that after the definition of the tumor volume as a planning target volume, a target point should be defined where the probe is positioned intraluminally or intratumorally [46]. In addition, the guidelines strongly suggest keeping a record of thermometry measurement points within or close to the tumor sites [43]. After completion of the HT session, recorded temperature data during t_treat_ are evaluated by computing temperature metrics. For instance, T_max_ is calculated as the maximum temperature value recorded in the target volume (Table 1). T_10_, another maximum temperature metric, is computed as the temperature value received by 10% of the target volume [32]. Similarly, the other temperature metrics listed in Table 1 are computed. In current practice, the thermometric parameters and thermal dose are computed by software integrated in the HT systems or using thermal analysis tools such as RHyThM [51].

To illustrate how temperature, t_pre_ and t_treat_ terms are measured in clinical practice, Figure 1 shows the temperature and heating duration parameters of a patient treated with HT in the radiation oncology center at Cantonal Hospital Aarau (KSA) using BSD-500 system (BSD Medical Corporation, Salt Lake City, UT, USA).

The temperature metrics and thermal doses can be also computed by using the data from Figure 1. A decade ago, a new thermal dose entitled “TRISE” was proposed by Franckena et al. [36]. However, this parameter has not yet been evaluated in experimental studies. Another newly proposed thermal dose parameter is the area under the curve (AUC) [49]. In contrast to CEM43°C and TRISE, AUC is computed without any prior assumptions by summating AUC for the entire treatment session, including t_pre_ and t_treat_. Similarly to TRISE, AUC has not yet been investigated in preclinical studies. Another parameter related to HT used in this review is thermal enhancement ratio (TER), defined as ‘the ratio between RT dose required to achieve a specific endpoint and RT dose to achieve the same endpoint in combination with HT’ [52]. 

## 3. Evidence for Predictive Values of Thermometric Parameters in Preclinical Studies

### 3.1. Heating Temperature 

The responsiveness of a tumor to HT is determined by different heat-induced mechanisms at the cellular level. The oxygenation rate is affected by temperature, as a higher rate was reported at 41–41.5 °C in comparison to higher temperature (at 43 °C) in rodent tumors, human tumor xenografts, canine, and human tumors [53]. Heating at 40 °C potentiated the cytotoxicity of CT drugs in human maxillary carcinoma cells [28], and the cytotoxicity was further increased on heating to 43.5–44 °C [54]. In contrast, another preclinical study showed no such dependency at 41–43.5 °C [55]. An in vitro study showed that apoptosis in human keratinocytes occurred at temperatures of 39 °C and above [56]. However, the majority of studies show synergistic actions of HT with RT and CT at temperatures above 41 °C [5,57], leading to the inhibition of DNA repair and chromosomal aberrations, induction of DNA breaks by RT and CT, and protein damage as an underlying molecular event of heat treatment [5,58,59]. To benefit from additive and synergistic effects of HT when combined with RT and/or CT, uniform temperature in the target volume should be delivered during the whole treatment course. 

The temperature metrics are used to present the heating temperature achieved during treatment, not only in the target volume, which encompasses the tumor, but also for adjacent healthy tissue. T_90_, T_80_, T_50_, T_20_, and T_10_ are considered to be less sensitive than T_min_, T_avg_ and T_max_, due to the number and arbitrary positioning of sensors in the tissue. Such temperature metrics can be used to understand the response to heat of various cancer types for a specific duration and, at the same time, the heat-induced effects on surrounding normal tissues. However, except for T_min_ and T_max_, most descriptive metrics of temperature have no specific reference values yet (Table 2).

T_50_ and T_90_ reference values are defined according to ESHO guidelines for treatment with superficial HT, but not for the deep HT technique. No reference values for temperature metrics are based on experimental data (Table 2), even though temperature distributions can be better controlled in preclinical than in clinical studies. In an in vivo study, no temperature variations were observed in tumors as they were recorded intratumorally [15]. Temperature at a reference value with minor variations (±0.05 °C) was reported in a vitro study [60]. In contrast, the temperature data recorded in patients are limited for various reasons. For example, thermistor probes inserted in deep-seated tumors in patients have the potential to cause complications or sometimes are impractical to insert intraluminally or intratumorally [61]. The value of the lowest temperature achieved during HT treatment is shown to have a prognostic role in describing the biological effects of HT. According to an in vivo study, T_90_ was a predictive parameter of reoxygenation and radiosensitization effects [62]. An in vitro experiment which investigated the difference in thermal sensitivity between hypoxic and oxic cells demonstrated that direct cytotoxicity induced by HT is more selective to the hypoxic cells [7]. Thus, temperatures required to achieve comparable thermal enhancement effect of HT vary depending on tissue type and characteristics. 

### 3.2. Heating Duration

Temperature fluctuations, such as a decrease by 0.5 °C, have been shown to have a strong effect on the extent of cell kill, which was compensated by doubling the heating duration [6,63]. Therapeutic ratio, defined as the ratio of thermosensitive liposomal doxorubicin delivered to the heated tumor increased from 1.9-fold with 10 min heating to 4.4-fold with 40 min heating [64]. In an in vivo study, TER for mouse mammary adenocarcinoma (C3H) increased with respect to heating exposure longer than 30 min at 41.5 °C [15]. A study used mouse leukemia, human cervical carcinoma (HeLa), and Chinese hamster ovary (CHO) cells to demonstrate that the time required to kill 90% of the cells at 43 °C varied according to type [65]. The survival data from different tissues were analyzed using the Arrhenius equation to understand the effect of t_treat_ for different cell types [66]. These analyses showed that the reference t_treat_ value is set at 60 min when heating constantly at reference temperature (Table 3). 

Heating for longer than 60 min is restricted by thermotolerance, which was observed after 20 min while heating at 43.5 °C [67]. In addition, the surviving fraction of asynchronous CHO cells heated to 41.5 °C was decreased with increasing t_treat_, until the thermotolerance effect appeared [21]. Thermotolerance is activated by different forms of stress including heat exposure for a specific time [68], which depends on the temperature and the amount of HT damage induced [69]. In an experimental study, the effect of thermotolerance was observed using the human tumor cell line (HTB-66) and CHO cells after 4 h of heating at 42.5 °C and 3 h of heating at either 42.5 or 43 °C [70]. The degree of thermotolerance is determined by cell type, heating temperature, and time of heating including the interval between successive heat treatments [71].

### 3.3. Thermal Dose 

The relationship between temperature and t_treat_ was demonstrated experimentally in two preclinical studies, which showed that the same thermal enhancement of ionizing radiation in cells lines was achieved by heating for 7–11 min at 45 °C or for 120 min at 42 °C [26,72]. It was also shown that different survival rates were obtained when heating asynchronous CHO cells to different temperatures for varying t_treat_ [66]. These preclinical data showed that heating temperature and t_treat_ influence thermal damage. The relationship of temperature and t_treat_ to the biological effects induced by HT is described using the Arrhenius equation, which models the relationship of the inactivation rate in a biological system [21]. This led to the discovery that the relationship between temperature and t_treat_ depends on the activation energy required to induce a particular HT-induced biological event, such as protein denaturation [59,66]. The thermal dose concept, CEM43°C, was established to account for the biological effects induced by HT in terms of both temperature and t_treat_ [21]. More specifically, CEM43°C calculates the equivalent time of a HT treatment session by correlating temperature, t_treat_ and inactivation rate of a biological effect induced by heat based on the Arrhenius equation. The reference temperature of 43 °C was shown as a breakpoint in the Arrhenius plot with a steeper slope between 41.5 and 43 °C in comparison to 43–57 °C [66]. The threshold values of CEM43°C for tissue damage differ for specific tissues as identified in in vivo studies and are reviewed elsewhere [70,73,74]. In addition, these data underline that CEM43°C is an important parameter that has biological validity to assess the thermal damage in tissues. CEM43°CT_90_ is one of the most frequently used thermal dose descriptors at T_90_, not only in clinical, but also in experimental settings. In an in vivo study, Thrall et al. [75] showed a relationship between CEM43°CT_90_ and local control in canine sarcomas, but not with CEM43°CT_50_ and CEM43°CT_10_. Another in vivo study using breast (MDA-MB-231) and pancreatic cancer (BxPC-3) xenografts showed that at relatively low values of CEM43°CT_90,_ tumor volumes could be reduced by exposure to heat alone [76]. However, none of the preclinical studies proposed reference values for clinical validation, as shown in Table 4. 

Although there is no reference threshold value for the CEM43°C, its efficacy to predict tumor response and local control has been experimentally proven [75,77]. CEM43°C is considered as a thermal dose parameter with few weaknesses which have been discussed elsewhere [78].

### 3.4. Number of HT Sessions

Thermotolerance is an undesirable side effect of HT which renders tumor cells insensitive to heat treatment for 48 to 72 h [79]. Thermotolerance consists of an induction phase, a development phase, and a decay phase. Each of these components may have its own temperature dependence as well as dependence on other factors, such as pH and presence of nutrients [80]. Thermotolerance plays an important role on how HT sessions are scheduled during the treatment course. An in vivo study using C3H mouse mammary carcinoma confirmed that preheating for 30 min at 43.5 °C induced thermotolerance for the next heating session [81]. Twice weekly heating to 43 °C for 60 min in combination with RT at 3 Gray (Gy) per fraction for 4 weeks was shown to result in a steady state decline in oxygenation level suggesting vascular thermotolerance [82]. In comparison, Nah et al. reported that heating at 42.5 °C for 60 min could render the tumor blood vessels resistant to the next heating session after an interval of 72 h [83]. It has also been shown that when HT was delivered daily with RT 5 days a week, no significant thermal enhancement could be detected in comparison to one single HT session, even when heat was delivered simultaneously or sequentially [84]. With the agreement of these findings, N_week_ is defined as 1 or 2 sessions separated by at least 72 h (Table 5).

In summary, HT should be delivered once or twice weekly, taking into account the type of cancer, RT fractionation and CT drug scheduling. Due to logistical reasons, the N_total_ usually depends on, the treatment plan for different cancer sites, number of RT fractions or number of CT cycles (Table 5).

### 3.5. Time Interval Parameter between HT and RT and/or CT

The t_int_ between HT and RT and/or CT treatment is another parameter that affects sensitization due to time-dependent biological effects and its contribution to thermotolerance. 

Recently, an in vitro study of human papillomavirus (HPV)-positive (HPV16^+^, HPV18+) and HPV-negative cell lines that were treated with HT either 0, 2 and 4 h before and after RT showed that the shortest t_int_ resulted in lower cell survival fractions and decreased DNA damage repair [85]. The influence of t_int_ has been investigated in an in vivo study, which reported that TER is greatest when heat and radiation are delivered simultaneously [15]. Unfortunately, simultaneous delivery is currently technically impossible in clinical routine and therefore heat and radiation are usually delivered sequentially. A very short t_int_ of approximately five min is considered as an almost simultaneous application [86]. Dewey et al. concluded that HT should be applied simultaneously or within 5–10 min either side of radiation to benefit maximally from the radiosensitizating effect of heat [6]. TER is decreased faster for the normal cells than for cancerous cells when t_int_ ≤ 4 h between HT and RT [15]. Thus, it can be argued that a slightly longer t_int_ could ensure the sparing of normal tissue from radiosensitization before or after RT. A t_int_ longer than 4 h, no sensitization effects induced by HT were observed [15,85]. The wide range of acceptable t_int_ values reported in experimental studies is from 0 (when CT is delivered during HT) to 4 h (Table 6). 

In contrast to RT, CT can be given simultaneously or immediately after or before HT [87]. A preclinical study, in which cisplatin and heat were used to treat C3H xenografts, showed that a higher additive effect can be obtained when cisplatin was given 15 min before HT in comparison with an interval longer than 4 h [55]. 

Furthermore, HT has been shown to sensitize the effects of gemcitabine at 43 °C when the drug was given 24 h after heating [88], whereas another study showed an optimal effect when the drug was given 24–48 h before heating [89]. The type of CT agent and its interaction with heat are factors which determine the t_int_ between HT and CT (Table 6). 

### 3.6. Sequencing of HT in Combination with and RT and/or CT

An additional predictive parameter for the effectiveness of radiosensitization and chemosensitization is the sequencing of heat prior to or after the application of RT or CT. Usually, HT and RT are delivered sequentially but there is no consensus as to the optimal sequence. An in vivo study by Overgaard investigated the impact of sequence and interval between the two modalities on local tumor control and normal tissue damage in a murine breast cancer model and found that the sequence did not have any significant effect on the thermal enhancement in tumor tissues [15]. However, an experimental study using Chinese hamster ovary (HA-1) and mouse mammary sarcoma (EMT-6) cell lines showed that sequencing of radiation and heat altered radiosensitivity for these two cancer cell types [90]. HT before RT showed more thermal enhancement in synchronous HA-1 cell lines and the opposite sequence increased the thermal enhancement in EMT-6 cell lines. Other experimental studies reported no impact of the sequence of RT and HT in V79 cells on thermal enhancement [26,72]. In line with these results, an experimental study with HPV cell lines showed no difference in radiosensitization or cell death when heat was delivered prior or after radiation [85]. Due to conflicting results with regard to the treatment sequencing of HT and RT, additional preclinical mechanistic studies on different cell types are required.

An in vivo study where heat was combined with cisplatin CT showed that simultaneous application of both treatments resulted in prolonged tumor growth delay in comparison with administration of cisplatin after HT [55]. Another study found that simultaneous exposure of human colorectal cancer (HCT116) cells to HT and doxorubicin was more effective than sequential administration because of higher intercellular drug concentrations at 42 °C [91]. In conclusion, better insight into the interaction of various CT drugs with HT and RT is required to define the optimal sequencing of specific drugs and RT dose.

## 4. Evidence for the Predictive Values of Thermometric Parameters in Clinical Studies Combining HT with RT

Numerous prospective and retrospective clinical studies have been conducted to assess the efficacy of HT in combination with RT for treating superficial and deep-seated tumors. The design of most clinical studies was based on the translation of experimental findings aiming to reproduce benefit of HT when combined with RT. 

Table 7 and Table 8 show the results of the most important clinical studies. The prospective clinical studies in Table 7 reported improved clinical results, apart from the study by Mitsumori et al. which did not show a significant difference in the primary clinical endpoint of local tumor control between two treatment arms [92]. The underlying reason could have been differences in RT dose prescriptions and missing patient treatment data. Although the study showed a significant difference in progression free survival, this was judged to be not a substantial benefit. The authors stressed the need for internationally standardized treatment protocols for the combination of HT and RT.

In reality, temperature and thermal dose are usually reported as post-treatment data recordings (Table 2 and Table 4) to account for temperature homogeneity or sensitivity. Even though temperature cannot always be measured invasively, depending on the location of the tumor, a strong correlation was reported between intratumoral and intraluminal temperatures, suggesting that intraluminal temperature measurements are a good surrogate for pelvic tumor measurements [50,93]. In addition, retrospective studies showed that a higher intra-esophageal temperature (>41 °C) predicts longer overall survival, improved local control and metastasis-free rate [94,95]. The difficulty of performing invasive measurements was illustrated by a randomized phase III study by Chi et al. [96] in which only 3 out of 29 patients with bone metastases had directly measured intratumoral temperature. In the study by Nishimura et al. [97], the HT session was defined as effective if an intratumoral temperature exceeded 42 °C for more than 20 min. However, according to the Arrhenius relationship, this is not considered long enough to induce a significant biological effect [21]. 

Another obstacle during HT is the non-standardized methodology for describing the temporal and spatial variance of temperature fields. Several groups have investigated the correlation between various temperature metrics. The study by Oleson et al. showed that T_min_, tumor volume, radiation dose, and heating technique play significant roles in predicting treatment response for patients treated with RT in combination with HT [29]. In contrast, Leopold et al. reported that the more robust parameters T_90_, T_50_, and T_10_ are better temperature descriptors and predictors of histopathologic outcome than T_min_ and T_max_ [32]. The median T_min_, T_min_ during the first heat treatment and tumor volume were reported to be factors predictive for the duration of cancer response (Table 7) [98], even though it is considered that skin surface temperature is not representative for superficial tumors and cannot be associated with clinical outcomes [42]. For deep-seated tumors, Tilly et al. reported that T_max_ was a predictive treatment parameter for prostate-specific antigen (PSA) control [99]. The relationship of high (T_avg_ ≥ 41.5 °C) and low (T_avg_ < 41.5 °C) tumor temperature with clinical response has been analyzed in a study by Masunaga et al. [100]. They showed that heating the tumor to temperatures of T_avg_ ≥ 41.5 °C for a duration of 15–40 min achieved better tumor down-staging and better tumor degeneration rates [100]. This finding supports the concept that direct cytotoxic effects of HT are enhanced at temperatures higher than 41 °C, as suggested in preclinical studies [5,57]. A higher response rate was also reported when tumors were heated with T_avg_ > 42 °C for 3–5 HT sessions [97]. In contrast, a study showed no difference in clinical outcome when patients were treated with mean T_min_ = 40.2 °C, T_max_ = 44.8 °C or T_avg_ = 42.5 °C for N_total_ of 2 or 6 [24]. Other studies also reported no impact of N_total_ and N_week_ on clinical outcome [40,101]. The contradictory results derived from clinical studies with regard to the predictive power of temperature descriptors and N_total_ are why we did not list reference values for these descriptors in Table 5. 

The predictive role of thermal dose has been investigated in both prospective and retrospective clinical studies (Table 7 and Table 8). However, there is still no conclusion about the values for thermal dose that should be obtained during HT treatment for maximal enhancement effect. In prospective studies (Table 7), the correlation between thermal dose and treatment outcome is rarely reported. Retrospective studies reported that thermal dose, CEM43°C, is an adequate predictor of treatment response and its best prognostic descriptor is CEM43°CT_90_ [32,33,36,37,38,102].

**Table 7 cancers-14-00625-t007:** Prospective clinical studies using RT in combination with HT.

Author(s)	Cancer Site, *n*	RT Dose (Gy) /Fractions	Temperature Metrics (°C)	HT Session	t_treat_ (min)	Thermal Dose (min)	t_int_ (min)	Sequence	Clinical Outcome (Comment)
Chi et al. [96]	Bone metastases, *n* = 29	30.0/10	T_max_ ^†^: 41.9 ± 1.2	N_total_: 4 N_week_: 2	40	n.r.	120	HT after RT	Increased 3-months radiologic CR ^1^ and PR ^2^ rate: 37.9% (11/29) and 66.7% (10/15), respectively. No grade III toxicity was reported. HT increased pain control rate, no progression of pain achieved after 29 days. (correlation of thermometric parameters with clinical outcome not presented)
Valdagni et al. [103]	Head & neck, *n* = 18	64.0–70.0 /32–35	T_max_ ^†^: 43.3 ± 0.2 T_min_ ^†^: 40.4 ± 0.2 T_50_: 41.8 ± 0.2 T_90_: 39.8 ± 0.02	N_total_: 6 N_week_: 2	n.r.	max CEM42.5°C ^5^: 83.84 ± 9.4 min CEM42.5°C: 12.8–2.1	20–25	HT after RT	3-month CR: 83.3% (15/18), PR: 5.56% (1/18), PD ^3^ rate of 11.1% (2/18), overall improved LC ^4^. 5-year nodal control rate: 68.6% with TER: 2.83. N_total_ of two or six yielded similar results (80% CR with 6 sessions vs. 87%, with 2 sessions). No enhanced acute or late toxicities were reported. Extensive thermal analysis performed: no relation between thermometric parameters and response or toxicity.
Jones et al. [35]	Superficial cancers, *n* = 56	30.0–66.0 /15–33 when previously unirradiated 60.0–70.0 /30–35	n.r.	N_total_: 4–10 N_week_: 2	60 min	CEM43°CT_90_ ^†^: 14.3 (0.57–36.21)	n.r.	n.r.	CR: 66.1%, LC for pre-irradiated tumors: 48%. CEM43°CT_90_ associated with CR rate. Greater than 10 CEM43°CT_90_ showed a significant LC benefit. The improvement in LC was most pronounced for patients who were previously irradiated. No significant toxicity or survival benefit was reported.
van der Zee et al. [104]	Locally advanced pelvic tumors, *n* = 182	Bladder: 66.0–70.0 /33–35 Cervix: 40.0–50.0 /23–28 with HDR-IRT ^23^ (^192^ Ir): 14.0 or LDR-IRT ^24^ (^192^ Ir): 20.0–30.0 Rectum: 46.0–50.0 /20–22	n.r.	N_total_: 5 N_week_: 1	60	n.r.	60–240	HT after RT	CR for all patients: 55%, bladder: 73%, cervical: 83%, rectal: 21%. 3-year LC for all patients: 38%, for bladder: 42%, for cervical: 61%, for rectal: 16%. 3-year OS ^6^ rate for all patients: 30%, for bladder: 28%, for cervical: 51%, for rectal: 22%. 2.2% had grade III-IV HT-related toxicity. (correlation of thermometric parameters with clinical outcome not presented)
Harima et al. [105]	Cervix cancer, *n* = 20	52.2/29 with HDR-IRT (^192^ Ir): 30.0/4	T_max_ ^†^: 41.8 ± 1.1 T_avg_ ^†^: 40.6 ± 1.0 T_min_ ^†^: 39.6 ± 0.9	N_total_: 3 N_week_: 1	60	n.r.	30	HT after RT	CR: 80% (16/20), PR: 15% (3/20), NC ^7^: 5% (1/20). 3-year local LRFS ^8^, DFS ^9^ and OS: 79.7%, 63.6% and 58.2%, respectively. Acute toxicity, grade III: 2 patients. Late toxicity, grade III: 1 patient. (correlation of thermometric parameters with clinical outcome not presented)
Mitsumori et al. [92]	Locally advanced non-small cell lung cancers, *n* = 40	66.0–70.0 /33–38	T_max_ ^†^: 41.3 (37.7–44.0) T_min_ ^†^: 39.5 (35.5–41.7) T_avg_ ^†^: 40.3 (37.0–42.7)	N_total_: 7 N_week_: 1	60	n.r.	n.r.	n.r.	RR ^10^: 45.0%. 1-year LRFS and OS: 67.5% and 43%, respectively. Acute toxicity, grade II: 4 patients and grade III: 2 patients. Late toxicity, grade II: 3 patients and no grade III. (correlation of thermometric parameters with clinical outcome not presented)
Masunaga et al. [100]	Urinary bladder cancer, *n* = 28	24.0/6	T_avg_ ^†^: 41.5 ± 1.1 (39–44)	N_total_: 4 N_week_: 2	15–40	n.r.	n.r.	HT after RT	T_avg_ ≥ 41.5 °C achieved better results: 83.3% (10/12) tumor down-staging and degeneration, 0% local recurrence, 33% distant metastasis, in contrast with T_avg_ < 41.5 °C. Survival rate was higher if T_avg_ ≥ 41.5 °C than T_avg_ < 41.5 °C. The toxicity associated with HT: pain during treatment.
Berdov et al. [106]	Advanced rectal cancer, *n* = 56	40.0/10	n.r.	N_total_: 4–5 N_week_: n.r.	60	n.r.	10	HT before RT	1-,2-,3-,4-, and 5-year survival: 61.8 ± 6.6%, 48.1 ± 6.7%, 43.9 ± 6.7%, 35.6 ± 6.4%, and 35.6 ± 6.4%. The mean for CR rate (>50%): 53.6% (30/56) and for CR rate (<50%): 23.3% (13/56). (correlation of thermometric parameters with clinical outcome not presented)
Maluta et al. [107]	Locally advanced high risk prostate cancer, *n* = 144	70.0–76.0 /35–38	Rectum T_max_ ^†^: 42.7 T_90_ ^†^: 40.2 (38.4–42.0) Bladder T_90_ ^†^: 41.3 (39.5–42.3)	N_total_: 4 N_week_: 1	n.r.	CEM40 °CT_90_ ^†^: 24.4 (14.4–34.4)	15–30	HT before RT	5-year OS: 87% and 5-year biochemical progression-free survival: 49%. No late grade III toxicity or significant acute HT-correlated toxicity. (correlation of thermometric parameters with clinical outcome not presented)
Leopold et al. [40]	Superficial cancers, *n* = 111	24.0–70.0 /7–28	n.r.	N_total_ ^†^:4.5(1–6) for N_week_=1 and 7 (2–13) for N_week_=2 N_week_: 1–2	60	n.r.	30–90	HT after RT	CR: 46%, PR: 34%, OS: 80%. T_90_ was significantly related to CR. Cumulative minutes of T_90_ ≥ 40 °C and logarithm of RT dose were predictive of both CR and OS. T_min_, N_week_, and N_total_ were not significantly related to either end points. Toxicity, grade IV: 1 patient and grade III: 7 patients.
Nishimura et al. [97]	Colorectal cancer, *n* = 33	40.0–70.0 /25–35	Abdominal wall & hip: T_max_ ^†^: 44.2 ± 2.1 T_avg_ ^†^: 42.6 ± 1.3 T_min_ ^†^: 40.5 ± 0.7 Perineum: T_max_ ^†^: 43.1 ± 1.7 T_avg_ ^†^: 42.2 ± 1.2 T_min_ ^†^: 40.5 ± 1.1 Pelvis: T_max_ ^†^: 42.1 ± 1.5 T_avg_ ^†^: 41.2 ± 1.5 T_min_ ^†^: 40.1 ± 1.1	N_total_: 2–14 N_week_: 1–2	40–60	n.r.	10–30	HT after RT	6- and 12-months LC: 59% (17/29) and 31% (8/21), respectively. CR rate: 11% (4/35) and PR: 43% (15/35). Better treatment response of unresectable colorectal cancers than recurrent tumors. Higher response rate of 67% reported when tumors heated with T_avg_ ^†^ > 42 °C for N_total_=3–5. N_total_ ≥ 5–14 showed not to increase the response rate.
Anscher et al. [108]	Prostate cancer, *n* = 21	65–70 /32–35	Intraprostate median T_90_ ^†^: 39.3 ± 0.9 T_50_ ^†^: 40.4 ± 0.8	N_total_: 5–10 N_week_: 1–2	60	CEM43°CT_90_ ^†^: 2.34 ± 3.23	60–154	HT after RT	Rectal temperatures were not predictive of prostate temperatures. The mean cumulative minutes with T_90_ of > 40 °C was 12 min in the prostate versus 28 min in the rectal lumen. 3-year DFS: 25% and 12 patients (67%) had relapsed. No higher complication of Grade III. T_90_, T_50_, and log(CEM43°CT_90_) were not significantly associated with time to relapse.
Gabriele et al. [109]	Inoperable or recurrent parotid carcinoma, *n* = 13	Inoperable: 70.0/35 Recurrent: 30.0/15	T_min_ ^†^: 40.28 ± 0.83 T_max_ ^†^: 42.83 ± 1.32	N_total_: 4–10 N_week_: 2	30–45	n.r.	n.r.	n.r.	CR: 80% (16/20), PR: 20% (4/20), LR ^11^: 20% (16/20), 5-year actuarial LC: 62.3 ± 13.2%. Higher maximum temperatures correlated with acute toxicity and maximum tumor diameter but without statistical significance. Major acute toxicities included three patients (15%) with superficial necrosis, 2/3 healed spontaneously within 4 to 6 months. No correlation between T_min_ and T_max_ and early or long term response was found.
Maguire et al. [110]	Soft tissue sarcomas, *n* = 35	50.0/25–27	n.r.	N_total_: 10 N_week_: 2	60	CEM43°CT_90_ ^‡^: 38 (0.1–601)	n.r.	n.r.	14% (5/35) of patients had non-heatable tumors. pCR ^12^: 52% (15/29), LF ^13^: 10% (3/29) with heatable tumors. DM ^14^: 14/30 patients with heatable tumors and 2/5 with non-heatable tumors. Thermal goal of CEM43°CT_90_ ≥ 10 reached for 25 out of 30 patients. Treatment–induced toxicity: 10/30 patients with heatable tumors. No correlation of thermal dose with histologic response was observed.
Tilly et al. [99]	Recurrent or locally advanced prostate cancer, *n* = 22	68.4/38	Primary cancer: T_90_ ^†^: 40.7 ± 0.3 T_max_ ^†^: 41.4 ± 0.4 Recurrent cancer: T_90_ ^†^: 40.6 ± 0.8 T_max_ ^†^: 41.0 ± 0.7	N_total_: 5–6 N_week_: 1	0–30	n.r.	30	HT before RT or HT after RT	6-year OS: 95% and 6-year RrR ^15^: 60%. Severe acute grade III toxicity: 8 patients and grade II: 2 patients. Late toxicity, grade III: 1 patient and grade II: 2 patients. No correlation between thermal parameters and toxicity. The thermal parameters were correlated with clinical endpoints: toxicity, PSA ^16^ control. T_max_ was the only relevant predictive factor for PSA control.
Lutgens et al. [111]	Locally advanced cervical cancer, *n* = 42	50.0/25 with HDR-IRT (^192^ Ir): 21.0/3 weekly or LDR: 32.0/1–2 or MDR: 29.0/1–2	n.r.	N_total_: 5 N_week_: 1	60	n.r.	60–240	HT after RT	Treatment failure in the pelvis: 19% (8/42). OS: comparable between RT + CT and RT + HT groups. Toxicity of grade ≥III: 5 patients. (correlation of thermometric parameters with clinical outcome not reported)
Hurwitz et al. [112]	Locally advanced prostate cancer, *n* = 37	66.60–70.0 /33–37	T_min_ ^†^: 40.1 (37.5–41.8) T_max_ ^†^: 42.5 (40.5–45.9) T_avg_ ^†^: 41.2 (39.2–42.8)	N_total_: 2 N_week_: 1	60	CEM43°CT_90_ ^†^: 8.4	60	HT before RT	7-year OS: 94% and failure free: 61%. 2-year DFS: 84% compared with a rate of 64% for similar patients on 4-month androgen suppression. The difference in median CEM43°CT_90_ between these patient groups who achieved 2.8 min and 10.5 min, respectively, was significant (*p* = 0.004). A small difference in DFS in favor of patients treated with higher temperatures.
Vernon et al. [113]	Localized superficial breast cancer, *n* = 306	DHG ^17^ (p): 32.0/16 DHG (r): 40.5–50.0/25 + boost: 10.0–20.0 MRC ^18^ BrR (p): 28.8/8 MRC BrI(r) + MRC BrR(r): 50.0/25 + boost: 15.0/5 ESHO ^19^: 32.0/8 PMH ^20^(p): 32.0/18 PMH(r): 50.0/25	DHG: T_90_ ^†^: 39.0 T_50_ ^†^: 40.7 T_max_ ^†^: 43.5 MRC BrR: T_90_ ^†^: 40.7 T_50_ ^†^: 42.5 T_max_ ^†^: 45.6 MRC BrI: T_90_ ^†^: 40.4 T_50_ ^†^: 42.3 T_max_ ^†^: 45.1 ESHO: T_90_ ^†^: 39.5 T_50_ ^†^: 41.1 T_max_ ^†^: 43.3 PMH: T_90_ ^†^: 40.7 T_50_ ^†^: 42.2 T_max_ ^†^: 44.6	n.r.	DHG: 60 (55–61) MRC BrR: 60 (30–60) MRC BrI: 60 (17–65) ESHO: 60 (60–60) PMH: 60	DGH: maximum of CEM42 °C ^†^: 0(0–69.5) CEM43°C ^†^: 3.95 (0–122) MRC: maximum of CEM42 °C ^†^: 9 (0–60) CEM43°C ^†^: 7.5 (0.1–87.7) ESHO: maximum of CEM42 °C ^†^: 5 (0–59) CEM43°C ^†^: 8.4 (0.2–74) PMH: maximum of CEM42 °C ^†^: 0 (0–32.8) CEM43°C ^†^: 1.5 (0–25) data from Sherar et al. [39]	n.r.	n.r.	Total CR: 59%, DHG: 73.6% (14/19), MRC BrI: 55.5% (10/18), MRC BrR: 56.67% (51/90), ESHO: 77.77% (21/27), PMH: 29.41% (5/17). CR rate of previously non-irradiated: 61% and CR rate of previously irradiated tumor: 46%. 2-year actuarial survival rate for all patients: 40%. Two largest studies (ESHO and MRC BrR) showed a statistically significant (*p* = 0.004 and 0.001, respectively) advantage for the addition of HT, whereas the other three trials do not show a benefit (ORs < 1). CR rate show dependency on size of tumor, the depth of the lesion, and on a history or presence of metastatic dis-ease outside the target area (univariate analysis). OS did not differ markedly but patients receiving HT has a marginally inferior survival. Sherar et al. [39]: initial CR rate is significantly correlated with thermal dose and no correlation between N_total_ and initial CR rate.
Datta et al. [114]	Head & neck cancer, *n* = 33	50.0 /25	n.r.	N_total_: 8–10 N_week_: 2	n.r.	n.r.	n.r.	HT before RT	RR: 76%, CR: 55%, PR: 21% and DFS: 33%. Particularly significant effect in patients with advanced disease. (correlation of thermometric parameters with clinical outcome not presented)
Overgaard et al. [115]	Recurrent or metastatic malignant melanoma, *n* = 63	24.0–27.0 /3	n.r.	N_total_: 3 N_week_: 1	60	CEM43°C ^†^: 9 (0–219) data from Overgaard et al. [116]	30	HT after RT	HT did not significantly increase acute or late radiation reactions. 5-year survival rate was 19% and was 38% for the patients for with control of all known disease. RR: 80%, initial CR rate: 62%, PR: 32%, NR: 20%, 2-year actuarial LC: 37%. The response rate was higher receiving 27 Gy than those receiving a lower dose. Both acute and late adverse effects were deemed acceptable. Overgaard et al. [116]: there is a significance of thermal dose relationship with the heat effect but no correlation between N_total_ and the outcome of treatment.
Dinges et al. [41]	Uterine cervix carcinomas, *n* = 18	50.4/28 with HDR-IRT (^192^ Ir): 20.0/4	T_20_ ^†^: 41.7 (40.3–43.2) T_50_ ^†^: 41.1 (39.2–42.5) T_90_ ^†^: 39.9 (37.7–41.9)	N_total_: 4 N_week_: 2	60	CEM43°CT_20_ ^†^: 48.2 (5.9–600.5) CEM43°CT_50_^†^: 15.2 (0.6–54.0) CEM43°CT_90_ ^†^: 6.8 (0.4–23.0)	n.r.	n.r.	CR: 13/18, PR: 4/18 and NR ^21^: 1/18. 2-year LC rate: 48.1%, development of distant metastases: 48.5% and DSS ^22^: 31.8%. CEM43°CT_90_ was a significant parameter in terms of local tumor control for N_tot_ = 4 (univariate analysis), but had no impact in terms of metastatic spread. T_20_, T_50_, T_90_, cumulative minutes of T_90_ > 40 °C, CEM43°CT_20_ and CEM43°CT_50_ were not significant in terms of local tumor control and DSS. No acute toxicity, grade III or IV. Late toxicity, grade III and IV: 3 patients.
Kim et al. [117]	Inoperable hepatoma, *n* = 30	30.6/17	n.r.	N_total_: 6 N_week_: 2	30–60	n.r.	30	n.r.	Subjective response rate: 78.6%, PR: 40%, stable disease: 46.7%, PD: 13.3%. 1-year survival values for all patients and for the partial responders were 34% and 50%, respectively. (correlation of thermometric parameters with clinical outcome not presented)
Engin et al. [98]	Superficial tumors, *n* = 50	60.0–70.0 /30–35 when previouslyirradiated: 40.0/10	Group A: T_min_ ^†^: 39.6 ± 0.2 Group B: T_min_ ^†^: 39.3 ± 0.2	Group A: N_total_: 4 N_week_: 1 Group A: N_total_: 8 N_week_: 2	60	Group A: CEM 43°C ^†^: 12.1 ± 3.9 Group B: CEM43°C ^†^: 15.0 ± 5.1	15–30	HT after RT	Group A patients treated with once-weekly HT session had CR: 59% (12/22), PR: 36% (8/22), NR: 5% (1/22). Group B patients treated with twice-weekly HT sessions had CR: 55% (12/22), PR: 45% (10/22). T_min_ did not influence response between Group A and Group B. Neither tumor response, duration of LC nor occurrence of skin reactions were significantly affected by N_week_.

*n*: number of patients assigned to be treated with HT in combination with RT; ^†^: mean value (±standard deviation) or mean value (range); ^‡^: median (range); n.r.: not reported; ^1^ CR: complete response; ^2^ PR: partial response; ^3^ PD: progressive disease; ^4^ LC: local control; ^5^ CEM42.5 °C: cumulative equivalent minutes at reference temperature 42.5 °C; ^6^ OS:overall survival, ^7^ NC: no change; ^8^ LRFS: local relapse-free survival; ^9^ DFS: disease free survival; ^10^ RR: responserate; ^11^ LR: local response; ^12^ pCR: pathological CR; ^13^ LF: local failure; ^14^ DM: distant metastasis; ^15^ RrR: recurrence rate; ^16^ PSA: prostate specific antigen; ^17^ DHG: Daniel den Hoed Cancer Center in Rotterdam; ^18^ MRC: Medical Research Council at the Hammersmith Hospital; ^19^ ESHO: European Society of Hyperthermic Oncology; ^20^ PMH: Princess Margaret Hospital/Ontario Cancer Institute; ^21^ NR: no response; ^22^ DSS: disease specific survival; ^23^ HDR-IRT: high dose rate interventional radiotherapy; ^24^ LDR-IRT: low dose rate interventional radiotherapy.

In a phase III study of the International Collaborative Hyperthermia Group, led by Vernon et al. [113], thermal dose was associated with complete response (CR) in patients treated for superficial recurrences of breast cancer [39]. Another randomized study showed that the best tumor control probability was dependent on thermal dose [106]. Further, retrospective analyses indicate that thermal dose is a significant predictor of different clinical endpoints (Table 8) [33,36]. A few studies did not find such significant relationships between clinical endpoints and thermal dose [103,109,110]. For example, in the prospective study of Maguire et al., a total CEM43°CT_90_ with a threshold above 10 min did not show a significant effect on CR [110]. However, the association of CEM43°CT_90_ with CR was later reported for patients treated with superficial malignant cancers [35]. Similar to the study by Maguire et al., the minimum effective thermal dose was set as 10 CEM43°CT_90._ In addition, a test HT session was performed to verify if the tumor was heatable, and a thermal dose of higher than 0.5 CEM43°CT_90_ could be achieved [35,110]. The objective of the study by Hurwitz et al. was to achieve a CEM 43 °CT_90_ of 10 min, yet the resulting mean of thermal dose for all 37 patients was only 8.4 min [112]. The cumulative minutes T_90_ > 40.5 °C, defined as ‘the time in minutes with T_90_ > 40.5 °C for the whole N_total_’, with a mean of 179 ± 92 min, together with T_90_ and T_max_ were reported to correlate with toxicity and prostate specific antigen clinical endpoints [99]. Similarly, Leopold et al. showed that cumulative minutes of T_90_ > 40 °C is a predictor of treatment endpoints [40]. In retrospective studies, TRISE thermal dose concepts [36] were shown to have a predictive role in treatment response. These retrospective analyses showed that TRISE had a significant effect on local control for a cohort of patients with cervical cancer [33].

The effect of the t_int_ parameter has been only analyzed with respect to treatment endpoints in retrospective studies. The study by van Leeuwen et al. reported that a t_int_ less than 79.2 min between RT and reaching 41 °C during HT was associated with a lower risk of in-field recurrences (IFR) and a better overall survival (OS) in comparison to a longer t_int_ [22]. In contrast, another retrospective study showed that neither a shorter t_int_ of 30–74 min nor a longer t_int_ of 75–220 min between RT and the start of HT were significant predictors of local control (LC), disease free survival (DFS), disease specific survival (DSS) or OS [33]. Thus, the optimal t_int_ between HT and RT to achieve a maximal effect on the tumor remains unknown.

Apart from heat-related parameters, the total dose of ionizing radiation and its fractionation in combination with HT has an impact on clinical treatment response [118,119]. Valdagni et al. [103] reported that increasing the total dose of RT appeared to improve clinical response as 71% (5/7) and 90% (9/10) CR rates were observed for patients with nodal metastases of head and neck cancers who received total doses of 64–66 Gy or 66.1–70 Gy, respectively. In addition, it was reported that previously irradiated tumors, which are typically more resistant to ionizing radiation, achieved higher CR rates when treated with combined RT and HT in comparison with RT alone [35].

Furthermore, RT technique has been reported to have a beneficial effect on combined RT and HT treatment outcomes [29]. For example, technological advance such as MRI-guided brachytherapy were shown to improve the treatment outcome when RT is combined with HT [36].

The weak, and in part contradictory, evidence from clinical studies clearly shows that further analyses of thermometric parameters are required to define reference values for clinical use. The reported values for thermometric parameters from prospective and retrospective clinical studies (Table 7 and Table 8) can be translated into standard references after being tested and validated in prospective clinical trials.

**Table 8 cancers-14-00625-t008:** Retrospective clinical studies using RT in combination with HT.

Author(s)	Cancer Site, *n*	RT Dose (Gy) /Fractions	Temperature Metrics (°C)	HT Session	t_treat_ (min)	Thermal Dose (min)	t_int_ (min)	Sequence	Clinical Outcome (Comment)
Franckena et al. [36]	Locally advanced cervix cancer, *n* = 420	46.0–50.4 /23–28 with HDR-IRT ^11^ (^192^ Ir): 17.0/2 weekly or LDR-IRT ^12^: 18.0/3 weekly or LDR: 30 Gy in 60 h	n.r.	N_total_: 5 N_week_: 1	60	CEM43°CT_90_ ^†^: 5.05 ± 4.18 min	n.r.	n.r.	CR ^1^: 78%, PR ^2^: 16%, SD ^3^: 3%, PD ^4^: 1%. 1-year PTC ^5^: 65% (95% CI: 60–70%), 5-year PTC: 53% (95% CI: 47–58%). 1 year DSS ^6^: 75% (95% CI: 71–79%) and 5-year DSS: 47% (95% CI: 41–53%). Toxicity of grade I: 51% (80/153), grade II: 39% (60/153), grade III: 9%(16/153) and grade IV: 0.6% (1/153). Tumor stage, performance status, radiotherapy dose and tumor size, CEM43°CT_90_ and TRISE emerged as significant predictors of the various end-points.
Kroesen et al. [33]	Locally advanced cervix cancer, *n* = 400	46.0 -50.4 /23–28 with HDR-IRT (^192^ Ir): 17.0/2 or MRI-IRT 7.0/3–4	n.r.	N_total_: 5 N_week_: 1	60	CEM43°CT_90_ ^†^: 3.40 (1.89–5.83) TRISE ^†^: 3.46 (2.93–3.86)	30–230	HT after RT	TRISE and CEM43T_90_ had a significant effect on LC (univariate and multivariate analyses). TRISE, and IGBT showed a significant effect on DFS ^7^, DSS, and OS ^8^ (univariate analyses). t_int_ grouped based on median value in short t_int_ (30–74 min) and long t_int_ (75–220 min) were not significant predictor of LC, DFS, DSS and OS. The incidence of late grade III toxicity did not differ between low or high TRISE or low or high t_int_ patients.
van Leeuwen et al. [22]	Locally advanced cervix cancer, *n* = 58	46.0–50.4 /23–28 with PDR: 24	n.r.	N_total_: 4–5 N_week_: 1	60	n.r.	33.8–125.2 ^†^	HT after RT	3-year IFR ^9^:18% (0–35%) in the short t_int_ (≤ 79.2 min) group and 53% (18–82%) in the long t_int_ (>79.2 min) group. 5-year OS: 52% (35–77%). OS ^‡^: 61 months (38–83 months) in the short t_int_ group and 19 months (13–26 months) in the long t_int_ group. No difference in toxicity was observed between short and long t_int_ group.
Franckena et al. [120]	Locally advanced cervix cancer, *n* = 378	46.0–50.4 /23–28 with HDR-IRT (^192^ Ir): 17.0/2 or 18.0–21.0/3 or 20.0–24.0/1 or HDR: 30.0/1	T_avg_ ^†^: 40.6	N_total_: 5 N_week_: 1	60	n.r.	30–240	HT after RT	CR: 77%. 5- year PTC: 53% for all patients (95% CI, 48–59) and 5-year DSS: 47% (95% CI, 41–53). N_total_ significant influence on CR, DSS and OS (univariate analysis) and on CR and DSS (multivariate analysis).
Oldenborg et al. [121]	Recurrent breast cancer, *n* = 78	32.0/8	T_90_ ^†^: 41.1 (37.7–42.4) T_50_ ^†^: 42.2 (39.0–43.4) T_10_ ^†^: 43.2 (41.0–44.5)	N_total_: 4 N_week_: 1	60	CEM43°CT_90_ ^†^: 22.3 (1.5–107.7) CEM43°CT_50_ ^†^: 37.3 (3.3–96.0)	60	HT after RT	3- and 5-year OS: 66% and 49%, respectively. 3- and 5-year LC: 78% and 65%, respectively. The only significant prognostic factor: time between primary and recurrent disease (multivariate analyses) CEM43°CT_90_ was not analyzed because skin temperature measurements are poor indicators of tumor temperature.
Datta et al. [49]	Muscle invasive bladder cancer, *n* = 18	unifocal cancer: 48.0/16 multifocal cancer: 50.0/20	T_avg_ ^†^: 40.5 ± 0.5 T_min_ ^†^: 36.7 ± 0.3 T_max_ ^†^: 42.0 ± 0.6	N_total_: 4 N_week_: 1	60	CEM43°C: 59.8 ± 45.6	15–20	HT before RT	16/21 patients were free from local recurrence until their last follow-up or death. Temperature attained during t_treat_ was significantly lower in patients with local failure. AUC > 37 °C and AUC ≥ 39 °C were significantly lower in patients who had a local relapse. N_week_ and N_total,_ no significant differences between CEM43°C and CEM43°C for T > 37 °C. T_avg_: significantly greater in patients with no local bladder failure for both individual and N_total_.
Leopold et al. [32]	Soft tissue sarcoma, *n* = 45	50.0–50.4 /25–28	T_90_ ^‡^: 39.5 T_50_ ^‡^: 41.6 T_10_ ^‡^: 43.0 T_min_ ^‡^: 37.7 T_max_ ^‡^: 44.0	Group A: N_total_: 5 N_week_: 1 Group B: N_total_: 10 N_week_: 2	60	n.r.	30–60	HT after RT	Strongest predictive value for cumulative minimum T_90_, average min T_90_, cumulative minutes of T_50_, and average minutes T_50_: 40.5 °C, 40.5 °C, 41.5 °C, and 41.5 °C, respectively. N_week_: 2 were superior to N_week_: 1 with respect to the degree of histopathologic changes, but not predictive. T_50_ and T_90_ are substantially temperature distribution descriptors.
Ohguri et al. [94]	Non-small cell lung cancer, *n* = 35	45.0–80.0 /23–30	T_max_ ^‡^: 43.2 (38.9–48.1) T_avg_ ^‡^: 42.6 (38.8–47.0) T_min_ ^‡^: 41.7 (38.6–45.6)	N_total_ ^‡^: 11 (3–17) N_week_: 1–2	40–70	n.r.	15	HT after RT	CR: 2%, PR: 66%, and NC ^10^: 14%. Median OS, local recurrence–free, and distant metastasis–free survival times: 14.1, 7.7, and 6.1 months, respectively. Acute toxicity: 14% and late toxicity: 17%. All thermal parameters, T_min_, T_avg_ and T_max_ of intraesophageal temperature significantly correlated with median radiofrequency-output power.

*n*: number of patients assigned to be treated with HT in combination with RT; ^†^: mean value (±standard deviation) or mean value (range); ^‡^: median (range); n.r.: not reported; ^1^ CR: complete response; ^2^ PR: partial response; ^3^ SD: stable disease; ^4^ PD: progressive disease; ^5^ PTC: pelvic tumor control; ^6^ DSS: disease specific survival; ^7^ DFS: disease free survival; ^8^ OS:overall survival; ^9^ IFR: in-field recurrence; ^10^ NC: no change; ^11^ HDR-IRT: high dose rate interventional radiotherapy; ^12^ LDR-IRT: low dose rate interventional radiotherapy.

## 5. Evidence for Predictive Values of Thermometric Parameters in Clinical Studies Combining HT and CT

The added value of combining CT with HT has been established, not only in in vitro and in vivo studies, but also in clinical studies. Randomized clinical studies, which demonstrate that the combination of CT and HT results in improved clinical outcome in comparison with single modality treatment [122,123,124,125], confirm the preclinical findings [126]. The positive prospective and retrospective clinical studies are summarized in Table 9 and Table 10 respectively, with a focus on thermometric parameters.

The effectiveness of CT drugs has been enhanced by HT in a variety of clinical situations, such as localized, irradiated, recurrent, and advanced cancers, but only few indications are really promising. Long term outcome data, e.g., regarding the combination of CT with HT for bladder cancer, underline the clinical efficacy of this treatment strategy [125]. Chemosensitization by HT is induced by specifics biological interactions between CT drugs and heat. The increased blood flow and the increased fluidity of the cytoplasmic membrane of the cells induced by HT increase the concentration of CT drugs within malignant tissues. Interestingly, Zagar et al. performed a joint analysis of two different clinical trials and reported no significant correlation between drug concentration and combined treatment effect of CT and HT [127]. However, only a few CT drugs with specific properties (Table 9 and Table 10) are good candidates to use with HT. Alkylating agents, nitrosureas, platinum drugs, and some antibiotic classes show synergism with HT, whereas only additive effects are reported with pyrimidine antagonists and vinca alkaloids [59]. For example, heat increases the cytotoxicity of cisplatin, as shown by in vitro and in vivo studies [28,55]. Cisplatin concentration increases linearly with temperatures above 38 °C when applied simultaneously [28,128]. Synergy between HT and CT could be obtained at temperatures below 43.5 °C in a preclinical study [55]. Similarly, enhanced toxicity has been demonstrated for bleomycin [126,129], liposomal doxorubicin [130], and mitomycin-C [131]. Based on the summary of preclinical data, van Rhoon et al. suggested a CEM43°C of 1–15 min from heating to 40–42 °C for 30–60 min for any free CT drug, including thermos-sensitive liposomal drugs [132].

Lower temperatures might increase the therapeutic window by differential chemosensitization of cancer and normal tissues. In the prospective study of Rietbroek et al. [133] in patients with recurrent cervical cancer treated with weekly cisplatin and HT, three temperature descriptors, T_20_, T_50_, and T_90_, including the time in minutes in which 50% of the measured tumor sites were above 41 °C, indicated a significant difference in these parameters between patients who did and who did not exhibit a CR after treatment. However, there was neither a difference in T_max_ between responders and non-responders in a cohort of patients with recurrent soft tissue sarcomas treated with CT and HT [134], nor in a cohort of patients with recurrent cervical cancer [135].

In a prospective study of patients treated with CT and HT for recurrent ovarian cancer, no significant relationship of T_90_ and T_50_ and CEM43°CT_90_ and CEM43°CT_50_ with clinical outcome was found [136]. Similarly, the independency of T_90_ and CEM43°CT_90_ was also demonstrated in a retrospective study in soft tissue sarcoma [137]. Although a relationship of thermal dose with treatment response has been reported by Vujaskovic et al. [138], the parameters CEM43°CT_50_ and CEM43°CT_90_ were not statistically different between patients who did or did not respond to the treatment. The low mean value of T_90_ =39.7 (33.5–39.8) °C reported in this study might be the reason for the non-significant relationship of thermal dose with the clinical endpoint in addition to other factors such as hypoxia and vascularization level of the tumor. The first randomized phase III study that assessed the safety and efficacy of CT in combination with HT also recorded a low (≤40 °C) mean value of T_90_ = 39.2 °C (38.5–39.8 °C). However, the thermometric data were not analyzed or reported in correlation with treatment response [123]. Further investigations are required to understand which temperature is needed to achieve a maximum therapeutic effect, according to the type of CT drug and its concentration.

**Table 9 cancers-14-00625-t009:** Prospective clinical studies using CT in combination with HT.

Author(s)	Cancer Site, *n*	CT Drug(s) (mg/m^2^) × Cycles	Temperature Metrics (°C)	HT Session	t_treat_ (min)	Thermal Dose	t_int_ (min)	Sequence	Clinical Outcome (Comment)
Issels et al. [123]	Localised high-risk soft-tissue sarcoma, *n* = 104	125 etoposide twice weekly × 4 1500 ifosfamide four times weekly × 4 50 doxorubicin once weekly × 4	T_max_ ^‡^: 41.8 (IQR: 41.1–43.2) T_20_ ^‡^: 40.8 (IQR: 40.1–42.3) T_50_ ^‡^: 40.3 (IQR: 39.5–41.0) T_90_ ^‡^: 39.2 (IQR: 38.5–39.8)	N_total_: 8 N_week_: 2	60	n.r.	n.r.	n.r.	The proportion of patients who underwent amputation was 6.7% (7/104). After surgery, 108 patients received mean dose of 53.3 ± 8.9 Gy. 2-year and 4-year LPFS ^1^: 58% (51–66%) and 42% (35–51%), respectively; 2-year and 4-year OS ^2^: 78% (72–84%) and 59% (51–67%), respectively; CR ^3^, PR^4^, SD ^5^, PD ^6^ rates were 2.5%, 26.3%, 55.9%, 6.8%, 8.5%, respectively. The most frequent nonhaematological adverse events, grade III or IV: 23 patients. (correlation of thermometric parameters with clinical outcome not presented)
Alvarez Secord et al. [136]	Refractory ovarian cancer, *n* = 30	40 doxil once weekly × 6	T_90_ ^†^: 39.78 ± 0.59 T_50_ ^†^:40.47 ± 0.56	N_total_: 6	60	CEM43°CT_90_ ^†^: 5.84 ± 5.66 CEM43°CT_50_ ^†^: 13.00 ± 11.25	0–60	HT after CT	PR: 10% (3/30), SD: 27% (8/30), PD: 63% (19/30). Median of PFS ^7^: 3.4 and OS: 10.8 months, respectively. Toxicity due to HT, grade III: one patient. No significant differences between the T_90_, T_50_, CEM43°CT_90_ or CEM43°CT_50_ and those patients who had PD compared to SD or PR. No significant change in overall QoL was found between baseline and after treatment.
Fiegl et al. [134]	Advanced soft tissue sarcoma, *n* = 20	1500 ifosfamide four times weekly × 7 100 carboplatin four times weekly × 7 150 etoposide four times weekly × 7	T_max_ ^†^: 40.6 (39.1–42.2)	N_total_: 8 N_week_: 2	60	n.r.	n.r.	n.r.	Time ^‡^ to progression: 6 and to OS: 14.6 months. 3- and 6-months PFR ^8^ estimates: 60% and 45%, respectively. Grade III/IV haematological toxicities during CT: 70%. Objective RR ^9^: 20% PR: 20% (4/20), PD: 45% (9/20); No difference in T_max_ between responders or non-responders.
Rietbroek et al. [133]	Irradiated recurrent cervical cancer, *n* = 23	50 cisplatin once weekly × 12	T_20_ ^†^: 41.9 ± 0.9 °C T_50_ ^†^: 41.3 ± 0.8 °C T_90_ ^†^: 40.5 ± 0.7 °C	N_total_: 12 N_week_: 1	60	n.r.	30	HT after CT	RR: 52% observed after a median number of 8 cycles of treatment. OS ^‡^ rate: 8 months, specifically for responders: 12 months. T_20_, T_50_, T_90_ values were higher for responders than non-responders but it did not show a statistical significance.
Zagar et al. [127]	Recurrent breast cancer, *n*_trial 1_ = 18 *n*_trial 2_ = 11	Trial A: 20–60 LTDL ^13^ every 21–35 days × 6 Trial B: 40–50 LTDL every 21–35 days × 6	max T_90_: 42.6 min T_90_: 36.0	N_total_: 6	60 min	n.r.	30–60	HT after CT	Combined trials (A and B), CR: 17.2% (5/29) and PR: 31% (9/29). Patients with at least one or 20% of HT sessions with a T_90_ of target below 39 °C had similar local objective RR. Toxicity, grade IV: three patients (10.3%) and grade III: six patients (20.7%). No drug dose response relationship was observed between trial A and B. (correlation of thermometric parameters with clinical outcome not presented)
Ishikawa et al. [139]	Locally advanced or metastatic pancreatic cancer, *n* = 18	1000 gemcitabine once weekly × 12	n.r.	N_total_: 20 N_week_: 1	40	n.r.	0–1440	HT before CT	Major grade III-IV adverse events were neutropenia and anemia, no sepsis. Objective RR: and disease control rates were 11.1% and 61.1%, respectively. OS ^‡^: 8 months, and the 1-year survival rate was 33.3%. (correlation of thermometric parameters with clinical outcome not presented)
Vujaskovic et al. [138]	Locally advanced breast cancer, *n* = 43	30–75 LTDL × 4 100–175 paclitaxel × 4	T_90_ ^†^: 39.7(37.7–41.8)	N_total_: 4 N_week_: 2	60	CEM43°CT_90_ ^†^: 11.5 (1.5–159.3)	60	HT after CT	CR: 9% (4/43) and pathological CR: 60% (26/43); 4-year DFS ^10^ and OS: 63% and 75%, respectively. CEM43°CT_90_ ^†^ in responders was significantly greater than non-responders, 28.6 and 10.3 min, respectively. Patients had grade III and IV toxicity No statistical difference in the CEM43°CT_50_ and CEM43°CT_90_ between treatment responders and non-responders.
de Wit et al. [135]	Recurrent uterine cervical carcinoma, *n* = 19	60, 70, 80 cisplatin once weekly × 6	T_max_ ^†^: 41.6 ± 0.7 (39.7–43.6)	N_total_: 6 N_week_: 1	60	n.r.	0	HT after CT	No dose limiting toxicity at the 80 mg/m^2^ dose level of cisplatin. CR: 1 patient (dose level 80 mg/m^2^), PR: 18 patients, SD: 18 patients and PD: 3 patients (dose level: 60–80 mg/m^2^) and OS ^‡^: 54%. The improvement rate in QoL ^11^: 82.5% No differences between responders and non-responders for tumor: contact temperatures, indicative temperatures, tumor volume, oral temperature increase or total power applied.
Sugimach et al. [124]	Oesophageal >carcinoma, *n* = 20	30 * bleomycin twice weekly × 3 50 * cisplatin once weekly × 3	n.r.	N_total_: 6 N_week_: 2	30	n.r.	n.r.	HT after CT	CR: 5%, PR: 25%, minimal response: 20%, NC ^12^: 50% and decrease of tumor size in comparison to CT treatment only. (correlation of thermometric parameters with clinical outcome not presented)

*n*: number of patients assigned to be treated with HT in combination withCT; ^†^: mean value (±standard deviation) or mean value (range); ^‡^: median (range); n.r.: not reported; *: in mg unit only; ^1^ LPFS: local progressionfree survival; ^2^ OS: overall survival; ^3^ CR: complete response; ^4^ PR: partial response; ^5^ SD: stable disease; ^6^ PD: progressive disease; ^7^ PFS: progression free survival; ^8^ PFR: progression free rate; ^9^ RR: response rate; ^10^ DFS: disease free survival; ^11^ QoL: quality of life; ^12^ NC: no change; ^13^ LTDL: low temperature liposomal doxorubicin.

**Table 10 cancers-14-00625-t010:** Retrospective clinical trial studies using CT in combination with HT.

Author(s)	Cancer Site, *n*	CT Drug(s) (mg/m^2^) × Cycles	Temperature Metrics (°C)	HT Session	t_treat_ (min)	Thermal Dose	t_int_ (min)	Sequence	Clinical Outcome (Comment)
Yang et al. [140]	Advanced non-small cell lung cancer, *n* = 48	1000 gemcitabine twice weekly × 6 75 cisplatin twice weekly × 6	n.r.	N_total_: 8 N_week_: 2	40–60	n.r.	n.r.	HT after CT or HT before CT	No CR ^1^ reported, PR ^2^: 37.5% (18/23), SD ^3^: 33.3% (16/23), PD ^4^: 29.2% (14/23). ORR ^5^: 37.5% and DCR ^6^: 70.8%. 1-and 2-year survival rates: 14% and 1.3%, respectively. Toxicity, grade III: 14 patients and grade IV: no patients. (correlation of thermometric parameters with clinical outcome not presented)
Tschoep-Lechner et al. [141]	Advanced pancreatic cancer, *n* = 23	1000 gemcitabine once weekly × 8 25 cisplatin twice weekly × 8	T_max_ ^†^: 42.1 (40.9–44.1)	N_week_: 2 N_total_ ^‡^: 8	60	n.r.	0	simultaneously	PR: 4.34% (1/23), SD: 30.4% (7/23), PD: 34.7% (8/23); OS ^7‡^: 12.9 months (CI: 9.9–15.9 months). Mild (grade 1 and 2) position-related pain during HT treatment. (correlation of thermometric parameters with clinical outcome not presented)
Stahl et al. [137]	Soft tissue sarcomas, *n* = 46	250 etoposide × 4 6000 ifosfamide × 4 50 adriamycin × 4	T_90_ ^†^: 39.90 ± 0.74 (good responders) and T_90_ ^†^: 39.42 ± 1.78 (bad responders)	N_week_: 2 N_total_ ^‡^: 8	60	CEM43°CT_90_^†^: 17.96 ± 7.16 (good responders) CEM43°CT_90_^†^: 11.07 ± 5.58 (good responders)	0	simultaneously	PR: 31.6% (6/19 in the good responder group for RECIST ^8^) to 37% (10/27 in the poor responder group for RECIST). SD: 63.2% (12/19 for the good responder group in WHO ^9^ and volume) to 70.3% (19/27 in the poor responder group for volume). T_90_ and CEM43°CT_90_ parameters did not differ significantly between the groups.

*n*: number of patients assigned to be treated with HT in combination withCT; ^†^: mean value (±standard deviation) or mean value (range); ^‡^: median (range); ^1^ CR: complete response; ^2^ PR: partial response; ^3^ SD: stable disease; ^4^ PD: progression disease; ^5^ ORR: objective response rate; ^6^ DCR: disease control rate; ^7^ OS: overall survival; ^8^ RECIST: Response Evaluation Criteria in Solid Tumors; ^9^ WHO: world health organization.

Based on preclinical studies, the delivery of simultaneous CT and HT is recommended to achieve the greatest chemosensitization effect by HT [55,142]. However, in contrast to experimental results [20,55], most of the prospective studies listed in Table 9 were designed to deliver heat sequentially, and in most studies the CT drugs were administered prior to HT. Despite the fact that a considerable supra-additive or synergistic effect can be achieved by the simultaneous delivery of CT and RT, the sequential application of CT and HT may protect normal tissues from chemosensitization. The cell killing of hypoxic and oxygenated tumor cells can still be obtained with sequential delivery of CT drugs and HT [54]. In clinical studies, the t_int_ between modalities is usually kept under an hour [122,127,133,136,138]. Of note, the study of Ishikawa et al. showed a different scheduling of gemcitabine and HT for the treatment of locally advanced or metastatic pancreatic cancer [139]. Patients enrolled in this clinical study were treated with HT prior to CT with a t_int_ of 0–24 h. This unique flexible relationship of gemcitabine cytotoxicity with the t_int_ and sequence was revealed in an in vitro study [143]. The specific properties of CT drugs are main factors in determining the most efficient treatment sequence between CT and HT for each class of drugs.

That treatment protocols might require individualized standards for HT thermometric parameters as has recently been illustrated by an interim analysis of cisplatin and etoposide given concurrently with HT for treatment of patients with esophageal carcinoma. This analysis showed a relationship between tumor location and temperature reporting, i.e., higher temperatures were achieved in distal tumors [144]. Similar treatment site-dependent analysis of thermometric parameters should be performed in future trials. Although the biology underlying the interaction between CT drugs and heat in cancer and normal tissues is largely unknown, thermometric parameters have been shown to predict outcome when HT is combined with CT. Therefore, as discussed above, no definitive conclusions can be drawn regarding the optimal thermometric parameters for an enhanced effect of HT with CT.

## 6. Evidence for Predictive Values of Thermometric Parameters in Clinical Studies Using RT and CT in Combination with HT

Clinical malignancies, in particular advanced and inoperable tumors, can be treated using triplet therapy consisting of CT, RT and HT as a maximal treatment approach. The number of prospective and retrospective clinical studies investigating this approach is limited, the most important of which are listed in Table 11 and Table 12, respectively. These studies have already reported the feasibility of this trimodal approach for cervical cancer, rectal cancer, and pancreatic cancer.

The optimal combination of CT, RT, and HT in a single framework is complex, be-cause so many biological processes underly the interactions between the three modalities. In addition, clinical factors often influence the optimal combination of RT and CT. A template with fundamental specifications for designing a clinical study with the trimodal treatment is proposed by Herman et al. [145].

Even though there is no consensus as to the optimal scheduling of trimodal treatment, clinical studies to date integrate HT in combination with daily RT and CT drugs based on the concept that CT should interact with both RT and HT. Scheduling CT weekly is most feasible in terms of maintaining an optimal t_int_ between HT sessions, drug administration, and RT fraction [145].

The reason why cisplatin is most frequently used in trimodality regimens is less based on a specific interaction with heat, but rather on extensive evidence from phase III randomized trials showing that cisplatin potently improves the antitumor efficacy of radiotherapy, albeit at the cost of increased toxicity. Drug concentration has been shown to affect treatment response [146], as proven experimentally [147]. A phase I-II study reported that a higher cisplatin dose (50 mg/m^2^) in comparison with a lower dose (20–40 mg/m^2^) combined with RT and HT was positively correlated with CR [146]. Interestingly, overall survival between patients treated with two different CT regimes in combination with RT and HT did not differ [148]. However, the study was limited by the small size of the patient cohort. With reference to Table 11, clinical studies using trimodality treatment usually used conventional fractionation schemes with 1.8–2.0 Gy per fractions, leaving it largely unknown whether other schedules such as hypofractionation (>10 Gy per week or large single fractions) might be biologically more favorable. The total dose varied according to cancer type. In the case of cervical cancer, brachytherapy at high dose rate (HDR) or low dose rate (LDR) was applied to deliver the boost dose [149,150]. Furthermore, high or low total RT dose was reported to have an influence on CR rate when combined with 5-FU, leucovorin and HT [151]. In contrast to CT and RT treatment parameters, HT treatment parameters were frequently not reported. Thermometric parameters, such as temperature and thermal dose including t_int_, are reported but not set as fixed treatment requirements as there are no accepted reference values.

Disregarding the Arrhenius relationship of heating temperature and t_treat_, Amichetti et al. [152] reported a short t_treat_ of 30 min with mean temperature range values of T_max_ = 43.2 °C (41.5–44.5 °C) and T_min_ = 40.1 °C (37–42 °C). This might explain why this study did not result in a higher CR rate in comparison to the previous study by Valdagni et al. [103]. A correlation of achieved temperature with treatment response such as disease-free interval to local relapse (DFILR) was reported in the study by Kouloulias et al. [153]. This study showed that the DFILR rate was greater in patients who achieved heating temperature T_90_ > 44 °C for longer than 16 min during HT treatment. No significant correlation of DFILR with mean values of temperature descriptor T_min_ was confirmed. Referring to the last row in Table 7, Table 8, Table 9, Table 10, Table 11 and Table 12, the clinical endpoints among studies differ, which adds another level of complexity to generalizing the thermometric parameter correlations reported in studies.

Thermal dose was reported less frequently than temperature measurements, hence there is a lack of information about its predictive role for treatment response. In one study, thermal dose was directly and proportionally associated with CR, as patients who exhibited CR after treatment with a measured CEM43°CT_90_ of 4.6 min in comparison with patients with a PR and a CEM43°CT_90_ of only 2.0 min [146]. Recently, a prospective phase II study investigating neoadjuvant triplet therapy in patients with rectal cancer showed that patients achieving good local tumor regression had received a high thermal dose [154]. However, no threshold, only the mean of CEM 43 °C, was reported. The retrospective analysis of thermometric parameters of the prospective study by Harima et al. [149] showed that >1 min CEM43°CT_90_ is the threshold value which significantly correlates with treatment response (CR and disease-free survival rates). It also confirmed that CEM43°CT_90_ below 1 min are insufficient to achieve enhancement of RT and CT [155]. Unfortunately, no further analyses of the relationship between HT treatment parameters with clinical outcomes in studies using triplet therapy were reported.

Furthermore, the optimal interval between heat, radiation and anticancer drugs is still unclear. With reference to preclinical and clinical outcomes, t_int_ affects the thermal enhancement effect of HT on both ionizing radiation and CT drugs. A particular interaction between HT and CT in terms of t_int_ was reported according to properties of the CT drugs. A short t_int_ between sequential HT and doxorubicin resulted in more rapid treatment response [153]. However, it is not clear whether the CT drug interacts primarily with RT only when administered on the same day or also during an extended time period. In the first scenario, CT and HT could typically be administered within a range of 1–6 h prior to RT to optimally exploit the biological interaction.

**Table 11 cancers-14-00625-t011:** Prospective clinical studies using RT and CT in combination with HT.

Author(s)	Cancer site, *n*	CT Drug(s) (mg/m^2^) × Cycles	RT Dose (Gy) /Fractions	Temperature Metrics (°C)	Session	t_treat_ (min)	Thermal Dose (min)	t_int_ (min)	Sequence	Clinical Outcome (Comment)
Amichetti et al. [152]	Locally advanced head & neck cancer, *n* = 18	20 cisplatin once weekly × 7	70.0/35	T_max_ ^†^: 43.2 (41.5–44.5) T_min_ ^†^: 40.1 (37–42) T_90_ ^†^: 40.4 (38.7–42.2)	N_total_: 2 N_week_: 2	30	CEM42.5 °C T_min_ ^†^: 4.36 (0–27) CEM42.5 °C T_max_ ^†^: 88 (31.8–174)	20	HT after RT & CT	CR ^1^: 72.2% (13/18), PR ^2^: 16.6% (3/18); NC ^3^: 11.1% (2/18). OS ^4^: 88.8%, 3-year actuarial survival and probability of remaining free of nodal disease: 50.3% and 53.3%, respectively. No temperature metrics correlated with an increased acute side effects and the amount of skin toxicity.
Maluta et al. [156]	Primary or recurrent locally advanced pancreatic cancer, *n* = 40	1000 gemcitabine × 1–2 30 cisplatin ×	30.0–66.0 /10–33	T_90_ ^†^: 40.5 (95% CI: 39.8–41) T_max_ ^†^: 41.1 (95% CI: 40.2–42.5)	N_total_: 3–10 N_week_: 2	60	n.r.	n.r.	CT before HT & RT	OS ^‡^: 15 (6–20) months The most common hematological toxicity was grade 2 anemia. Toxicity, grade III: 5 patients. (correlation of thermometric parameters with clinical outcome not presented)
Asao et al. [151]	Locally advanced rectal cancer, *n* = 29	250 5-fluorouracil for 5 days × 2 25 for 5 days × 2	40.0–50.0 /20–25	T_max_ ^†^: 40.3 ± 0.89 (38.6–41.9)	N_total_: 3 N_week_: 1	60	n.r.	n.r.	HT after RT during CT	Toxicity, grade III: 2 patients. CR: 55.5% in patients with a total radiation dose of 50 Gy, which was significantly higher compared to patients treated with 40 Gy. 41.4% of patients had significant downstaging. (correlation of thermometric parameters with clinical outcome not reported)
Westermann et al. [150]	Cervix cancer, *n* = 68	40 cisplatin once weekly × 35	45.0–50.4 /25–28 with LDR- IRT ^7^ and HDR-IRT ^7^ (^192^ Ir)	T_90_ ^†^: 39.4 T_50_ ^†^: 40.7	N_total_: 8–10 N_week_: 1	60	n.r.	n.r.	HT & CT after/before RT	CR: 90%, 2-year DFS ^5^ and OS: 71.6% and 78.5%, respectively. A significant difference in DFS between Netherlands and US clinical centers. Specific toxicity associated with HT was mild. (correlation of thermometric parameters with clinical outcome not reported)
Harima et al. [149]	Locally advanced cervical cancer, *n* = 51	30–40 cisplatin once weekly × 3–5	30.0–50.0 /15–25 with LDR- IRT7 (192 Ir): 5.0–6.0 /3–5	T_max_ ^†^: 42.2 (40.1–44.6) T_avg_^†^: 41.1 (39.6–42.5) Data from Ohguri et al. [155] T_90_ ^‡^: 38.9 (37.7–42.2) T_50_ ^‡^: 39.9 (38.4–42.4)	N_total_: 4–6 N_week_: 1	60	CEM43°CT_90_ ^†^: 3.8 (0.1–46.6)	20	HT after RT&CT	CR: 88% (44/50). 5-year OS, DFS, and LPFS ^6^ were 77.8%, 70.8% and 80.1%, respectively. It was well tolerated and caused no additional acute or long term toxicity. Ohguri et al. [155]: CEM43°CT_90_ ≥ 1 min tended to predict better DFS and CR.
Kouloulias et al. [153]	Recurrent breast cancer, *n* = 15	40–60 liposomal doxorubicin once monthly × 6	30.6/17	T_max_ ^†^: 43.2 (41.5–44.5) T_min_ ^†^: 45.0 (44.2–45.7)	N_total_: 6 N_monthly_: 1	60	n.r.	180–240	HT after CT&RT	CR: 2% (3/15), PR: 80% (12/15); CR or PR obtained more quickly with a shorter t_int_ between HT and CT. DFILR ^7^ was better for T_90_ > 44 °C of ≥16 min compared with those for whom T_90_ > 44 °C of <16 min. DFILR was significantly correlated with T_min_ ^†^ but not with T_max_ ^†^.
Herman et al. [146]	Locally advanced malignancies, *n* = 24	20–50 cisplatin once weekly × 6	60.0–66.0 /30–33 or 24.0- 36.0 /12–18	T_max_ ^†^: 43.7 ± 2.6 T_min_ ^†^: 38.2 ± 2.0 T_avg_ ^†^: 40.8 ± 1.9	N_total_: 6 N_week_: 1	60	CEM42 °CT_90_ ^†^: 11.2 ± 21.3 CEM43°CT_90_ ^†^: 3.1 ± 5.4	n.r.	HT before CT&RT	CR: 50% (12/24), PR: 50% (12/24); No grade IIII acute toxicity. Late toxicity, grade IV: only 1 patient. With thermal dose of CEM43°CT_90_ ^†^ = 4.6 min, 50% of patients achieved CR and with CEM43°CT_90_ ^†^ = 2.0 min, 50% patients achieved PR. Cisplatin concentration amount correlated with CR.
Barsukov et al. [157]	Locally advanced rectal cancer, *n* = 68	650 capecitabine on days 1–22 × 6–8 50 oxaliplatin on days 3, 10 and 17 after × 6–8 10 metronidazole on days 8 and 15	40.0/10	n.r.	N_total_: 4 N_week_: 2	60	n.r.	60	n.r.	2-year OS: 91%, DFS: 83% and local RR: 13.6% R0 resection was achieved in 59 (92.2%). only five (7.8%) untreated patients remained inoperable. 12 (18.7%) and 1 (1.6%) patients had grade III and IV toxicity, respectively. (correlation of thermometric parameters with clinical outcome not presented)
Ott et al. [158]	Locally advanced or recurrent rectal cancer, *n* = 105	250 5-fluorouracil on days 1–14 and 22–35 or 1650 capecitabine on days 1–14 and 22–35 50 oxaliplatin × 4	LARC 50.4/28 LCC 45/25	n.r.	N_total_ ^‡^: 10 N_week_: 2	60	LARC ^19^ CEM43°C ^†^: 6.4 ± 5.2 LCC ^20^ CEM43°C ^†^: 6.4 ± 4.9	n.r.	HT before RT	11% (2/19) and 27% (16/59) DLT ^8^ criteria, corresponding to FR ^9^: 90% and 73%, respectively. Pathological CR: 20% (19/95), CTR ^10^: 28% (18/64) and 38% (3/8) in patients with LARC and LRRC, respectively. 5-year OS: 75% for the whole group. No grade 4–5 adverse events. (correlation of thermometric parameters with clinical outcome not presented)
Gani et al. [154]	Locally advanced rectal cancer, *n* = 78	1000 5-fluorouracil × 4	50.4/28	T_90_ ^‡^: 39.5 (IQR: 39.1–39.9)	N_total_: 8 N_week_: 2	60	CEM43°C ^‡^: 4.5 (IQR: 2.2–8.2)	n.r.	n.r.	19/78 (24%) patients: died or had tumor recurrence. 3-year OS: 94%, DFS: 81%, LC ^11^: 96% and DC: 87%. Pathological CR: 14% (the threshold not met). Patients with good tumor regression had higher values for CEM43°C. Comparable global health status with the data from general population based on EORTC-QLQ-C30 ^12^.
Rau et al. [159]	Locally advanced rectal cancer, *n* = 37	300–350 5-fluorouracil 50 * mg leucovorin 5 times weekly × 2	45.0–50.0 /25	Data from Rau et al. [160]: T_90_ ^†^: 40.2 ± 1.2 T_max_ ^†^: 41.4 ± 0.6	N_total_ ^‡^: 5 N_week_: 1	60	Data from Rau et al. [160] CEM43°CT_90_ ^†^: 7.7 ± 5.6 CEM43°CT_max_ ^†^: 33.1 ± 28.0	n.r.	RT after concurrent HT&CT	Grade III toxicity: 16%. ORR ^13^: 89%, and 31 resection specimens had negative margins. RR ^14^: 59.4%, CR: 14%, OS: 56%. Cumulative minutes at T_90_ ≥ 40.5 °C and T_90_ correlate with the RR but not with long term OS and DFSR ^15^ [160] but T_max_ showed no significant influence on RR. RR: 33% when T_90_ < 40.5 °C and RR: 75% response, T_90_ > 40.5 °C.
Wittlinger et al. [161]	Bladder cancer, *n* = 45	20 cisplatin 5 times weekly × 2 600 5-fluorouracil 5 times weekly × 2	50.4–55.8/ 28–31	T_avg_ ^†^: 40.8 (95%CI: 40.5–41.6)	N_total_: 5–7 N_week_: 1	60	CEM43°C ^†^:57 (95%CI: 40.5–41.6)	60	RT after concurrent CT&HT	CR: 96%, NC: 4%. Freedom from any local and distant relapse: 69% and relapse: 16%. 3-year bladder preservation: 96%, LPFS: 81%, DSS: 88%, DFS: 71%, OS: 80% and MFS ^16^: 89%. One of significant prognostic factors for OS: N_week_. Acute toxicity, grades III–IV: 27%. Late toxicity, grades III-IV: 24%.
Milani et al. [162]	Recurrent rectal cancer, *n* = 24	350 5-fluorouracil 5 times weekly × 4 (continuous infusion)	30.0–45.0/ 16–25	T_90_ ^†^: 41.4 T_50_ ^†^: 42.9 T_20_ ^†^: 43.5	N_total_ ^‡^: 8 N_week_: 2	60	n.r.	60	HT after concurrent RT&CT	CR: 0% (0/20), PR: 10% (2/20), NC: 85% (17/20), PD: 5% (1/20). 1-year OS, DMFS ^17^, LPFR ^18^: 87%, 82%, 61%, respectively. 2-year OS, DMFS, LPFR: 60%, 52%, 30%, respectively. 3-year OS, DMFS, LPFR: 30%, 39%, 15%, respectively. Acute toxicity, grade III: 12.5% of the patients. (correlation of thermometric parameters with clinical outcome not presented)

*n*: number of patients assigned to be treated with HT in combination with RT and CT; ^†^: mean value (±standard deviation) or mean value (range); ^‡^: median (range); ^1^ CR: complete response; ^2^ PR: partial response; ^3^ NC: no change; ^4^ OS: overall survival, ^5^ DFS: disease free survival; ^6^ LPFS: local progression free survival; ^7^ DFILR: disease-free interval to local relapse; ^8^ DLT: dose limiting toxicities; ^9^ FR: feasibility rate; ^10^ CTR: complete tumor regression; ^11^ LC: local control; ^12^ EORTC-QLQ: European Organization for research and treatment of cancer-quality of life questionnaire; ^13^ ORR: objective response rate; ^14^ RR: response rate; ^15^ DFSR: disease-free survival rate; ^16^ MFS: metastasis-free survival; ^17^ DMFS: distant metastases-free survival; ^18^ LPFR: local progression-free survival; ^19^ LARC: locally advanced rectal cancer; ^20^ LCC: recurrent rectal cancer.

**Table 12 cancers-14-00625-t012:** Retrospective clinical studies using RT and CT in combination with HT.

Author(s)	Cancer Site, *n*	CT Drug (s)(mg/m^2^) × Cycles	RT Dose (Gy) /Fractions	Temperature Metrics (°C)	Session	t_treat_ (min)	Thermal Dose (min)	t_int_ (min)	Sequence	Clinical Outcome (Comment)
Zhu et al. [163]	Locally advanced esophageal cancer, *n* = 78	450 5-fluorouracil five times weekly × 4–6 25 cisplatin five times weekly × 4–6 6	60.0–66.0 /30–33	n.r.	N_total_: 6–12 N_week_: 2	60	n.r.	120	n.r.	CR ^1^: 39.7% (31/78), PR ^2^: 56.4% (43/78), SD ^3^: 3.9% (3/78). 1-, 2- and 3-year LRC ^4^: 76.9%, 55.1% and 47.4%, respectively; 1-, 2- and 3-year DMFS ^5^: 67.9%, 38.5% and 30.8% respectively; 1-, 2- and 3-year OS ^6^: 67.9%, 41.0% and 33.3%, respectively (correlation of thermometric parameters with clinical outcome not presented)
Ohguri et al. [148]	Locally advanced pancreatic cancer, *n* = 20	Group A: 40–50 gemcitabine twice weekly × 4 Group B: 200–500 gemcitabine once weekly × 3	50.4–64.8 /28–36	n.r.	N_total_: 6 N_week_: 1	n.r.	n.r.	Group A:Instant Group B: 60–180	HT after CT&RT	Grade II-IV hematological toxicities: 8 patients. The objective tumor response, CR for 1 patient, PR for 4, and NC ^7^ for 15. DM ^8^: 13 and LF ^9^: 5 patients. DPFS ^10^: 8.8 months, OS ^‡^: 18.6 months. The treatment regimen did not correlate with the survival rates. (correlation of thermometric parameters with clinical outcome not presented)
Gani et al. [164]	Locally advanced rectal cancer, *n* = 60	1000 5-fluorouracil × 4	50.4/28	T_90_ ^‡^: 39.3 (37.1–40.6)	N_total_ ^‡^: 4 N_week_: 1–2	60	CEM43°C ^‡^: 1.1 (0.0–9.2)	n.r.	n.r.	5-year OS, DFS ^11^, local control and DMFS were 83%, 75%, 93% and 76%, respectively. No impact of HT on DFS and DMFS. N_total_ not predictive for OS, DFS, LC, or DMFS. Postoperative nodal stage remained a significant prognosticator for OS, DFS and DMFS (multivariate analysis).
Merten et al. [165]	Bladder cancer, *n* = 79	20 cisplatin 5 times weekly × 2 600 5-fluorouracil 5 times weekly × 2	50.4–55.8/ 28–31	n.r.	N_total_: 5–7 N_week_: 1	60	n.r.	0–60	RT after concurrent CT&HT	CR: 87% (67/77). 5- and 10-year OS: 87% and 60%, respectively. 5- and 10-year DFS to 66% and 46% respectively. Acute toxicity, grade III: 11% and grade IV: 3%. Late toxicity, grade III: 1.3%. (correlation of thermometric parameters with clinical outcome not presented)
van Haaren et al. [166]	Esophageal cancer, *n* = 29	50 paclitaxelonce weekly × 5 and carboplatin (AUC=2) once weekly × 5	41.4/23	T_90_ ^†^: 38.6 ± 0.5 T_50_ ^†^: 39.2 ± 0.6 T_10_ ^†^: 40.1 ± 0.8	N_total_: 5 N_week_: 1	60	n.r.	0–60	HT after CT & RT	CR: 19% (5/29), mPR ^12^: 26% (7/29), PR: 33% (9/29) and SD: 22% (6/29). The dependence of T_50_ on the body size parameters was substantial. (correlation of thermometric parameters with clinical outcome not presented)

*n*: number of patients assigned to be treated with HT in combination with RT; ^†^: mean value (±standard deviation) or mean value (range); ^‡^: median (range); ^1^ CR: complete response; ^2^ PR: partial response; ^3^ SD: stable disease; ^4^ LRC:locoregional control, ^5^ DMFS: distant metastasis-free survival; ^6^ OS: overall survival; ^7^ NC: no change; ^8^ DM: distant metastases; ^9^ LF: local failure; ^10^ DPFS: disease progression-free survival; ^11^ DFS:disease free survival; ^12^ mPR: partial remission with only residual microscopic tumor foci.

Moreover, the N_total_ was shown to be a prognostic factor for OS for bladder cancer patients treated with combined CT, RT, and HT followed by surgery [161]. In contrast, Gani et al. [164] reported that the number of HT sessions was not predictive for OS, DFS, LC, or distant metastasis-free survival. Neither did the sequencing of CT, HT, and RT in clinical reports follow a specific pattern. Preclinical studies are required to better understand the interaction of CT, RT, and heat and how they should be combined in future clinical trials.

## 7. Future Prospects

The main limitations of HT as a cancer treatment in current clinical practice are the need for better standardization of treatment protocols, up-to-date quality assurance guidelines that are widely applicable and dedicated planning systems to generate patient treatment plans. The wide variation of thermometric parameters derived from clinical studies indicate that HT treatment is currently delivered according to individual clinical center guidelines. Consequently, the comparison of clinical study outcomes is substantially hampered by the large degree of variation in treatment parameters. Regarding the data summarized in Table 6, Table 7, Table 8, Table 9, Table 10 and Table 11, apart from thermal dose and temperature measured during treatment, other thermometric parameters reported often include only t_treat_, t_int_, or N_week_.

Monitoring and measuring temperature is one of the main challenges in routine clinical practice and has hindered the clinical expansion of HT. The future of HT in combination with RT and CT requires novel technical developments for the delivery and measurement of homogenous heating of the malignant tissues. Not all studies (Table 7, Table 8, Table 9, Table 10, Table 11 and Table 12) recorded temperatures in the region of the tumor. The process of inserting temperature probes to monitor and record the HT is considered invasive and uncomfortable, and sometimes the tumor site is inaccessible for the temperature probe. For example, Milani et al. [162] reported that even though the tumors were not deep-seated, intratumoral temperature measurements were only feasible in one of 24 patients, so no representative thermal doses could be reported. One of the non-invasive approaches currently under clinical evaluation is magnetic resonance thermometry (MRT) that provides 3-D temperature measurements. Hybrid MR/HT devices are currently installed in five European clinical centers.

Temperature measurements in anthropomorphic phantoms with MRT are accurate in comparison with thermistor probes [167], but clinical measurements are currently inaccurate in most pelvic and abdominal tumors [168]. The physiological changes in tissue microenvironment, patient movements, magnetic field drift over time, limited sensitivity in fatty tissues, and respiratory motion, including cardiac activity in regions of the pelvis and abdomen, hamper the accurate temperature measurement by MRT [168]. The temperature images from MRT systems contain image distortion, artifacts, and noise, leading to inaccurate temperature measurement, low temporal resolution, and low imaging to signal-to-noise ratio (SNR) [169]. The sources and solutions of image artifacts as a result of additional frequencies were described by Gellermann et al. [170]. Proton-resonance frequency shift (PRFS), apparent diffusion coefficient (ADC), longitudinal relaxation time (T_1_), transversal relaxation time (T2), and equilibrium magnetization (M0) are the imaging techniques used to exploit temperature-dependent parameters [170,171,172,173]. The PRFS technique is the most frequently used MRT method, even though it was shown that when there is a poor magnetic field homogeneity, ADC or T_1_ techniques are preferable [174]. However, the accuracy of temperature measurements was in the range of ±0.4 to ±0.5 °C between PRFS method and thermistor probe using a heterogeneous phantom [175]. A stronger correlation between MRT and thermistor probes was found in patients with soft tissue sarcomas of lower extremities and pelvis [176] in comparison with recurrent rectal carcinoma [177]. The successful implementation of MRT in clinical centers, as automated temperature feedback during the HT session, might have a considerable impact on clinical outcomes to deliver the desired heating and conform the heat distribution to spare healthy surrounding tissues. This could substantially help to standardize data collection and the analysis of thermometric parameters. Another experimental approach to monitoring treatment temperature during HT sessions is electrical impedance tomography (EIT) as recently reported in a simulation study by Poni et al. [178]. EIT captures the electrical conductivity of tissues depends on temperature elevation. For example, the multifrequency EIT technique detects the changes in conductivity due to perfusion increase induced by the change in temperature [179]. The accuracy of EIT for temperature measurements was reported to range from 1.5 °C to 5 °C [180]. The potential of EIT to monitor temperature in the cardiac thermal ablation field is being investigated [181]. This technique also holds promise for HT treatment. Both MRT and EIT may allow for improvement of the spatial homogeneity of heat to the cancer tissues.

The technological advances and standardization of international treatment protocols for different cancer types will improve the effectiveness and synergy of HT in combination with RT and/or CT. In line with this, there is a need for clinically accepted processes for the recording and reporting of thermometric data. This will allow for the inclusion of specific thermometric parameters in future clinical studies combining HT with RT and/or CT. For any future prospective study, it should be mandatory that thermometric parameters are recorded and some recommendations are available in the current guidelines [43,46]. The integration of thermometric parameters is one of the objectives of the HYPERBOOST (“Hyperthermia boosting the effect of Radiotherapy”) international consortium within the European Horizon 2020 Program MSCA-ITN. The HYPERBOOST network aims to create a novel treatment planning system, including the standardization of thermometric parameters derived from retrospective and prospective clinical trials.

## 8. Conclusions

In this review, we provide an extensive overview of thermometric parameters reported in prospective and retrospective clinical studies which applied HT in combination with RT and/or CT and their correlation with clinical outcome. It is recognized that there is a wide variety in the practice of HT between clinical centers, and we aimed to elucidate the use and reporting of thermometric parameters in different clinical settings. It emerged that the sequencing of HT and RT varies more than the sequencing of HT and CT. Only a few standards seem to exist with regard to the sequence of HT with RT and CT in a triplet for specific CT drug, RT fractionation and thermal dose. According to the evaluated studies, t_int_ is a critical parameter in clinical routine, but no clinical reference values have been established. Of note, a constant t_treat_ of 60 min throughout the HT treatment course was described in most clinical studies. The most important parameter seems to be temperature itself, which correlates with thermal dose. Revealing the relationship between thermal dose and treatment response for different cancer entities in future clinical studies will lead to the improved application of heat to promote the synergistic actions of HT with RT and CT. We suggest that it become mandatory for new clinical study protocols to include the extensive recording and analysis of thermometric parameters for their validation and overall standardization of HT. This would allow for the definition of thermometric parameters, in particular of thresholds for temperature descriptors and thermal dose.

## Figures and Tables

**Figure 1 cancers-14-00625-f001:**
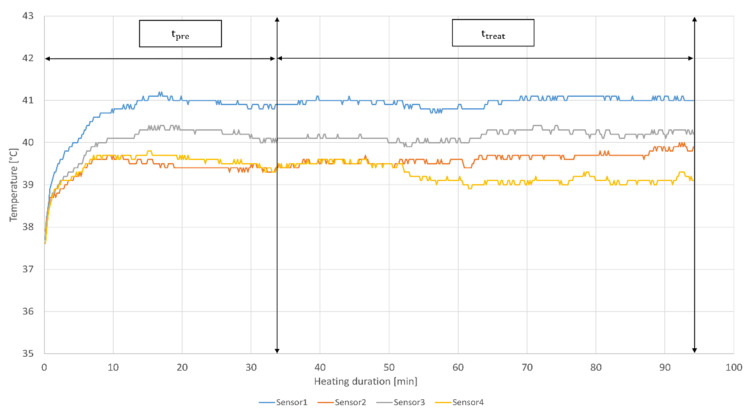
Recorded treatment data of a single HT session for a breast cancer patient. Temperature in °C and heating duration in minutes are measured non-invasively using four sensors located in close proximity to the tumor tissue. t_pre_ and t_treat_ of 33 and 60 min respectively according to the KSA clinical protocol are indicated.

**Table 1 cancers-14-00625-t001:** Definition of thermometric parameters.

Thermometric Parameters	Definitions
Heating Temperature	
T_min_	Minimum temperature achieved in target volume (°C).
T_max_	Maximum temperature achieved in target volume (°C).
T_avg_	Average temperature achieved in target volume (°C).
T_10_	Temperature achieved in 10% of the target volume (°C).
T_20_	Temperature achieved in 20% of the target volume (°C).
T_50_	Temperature achieved in 50% of the target volume (°C).
T_80_	Temperature achieved in 80% of the target volume (°C).
T_90_	Temperature achieved in 90% of the target volume (°C).
Heating duration	
t_pre_	Warm-up period is the time required to achieve the desired treatment temperature and therapeutic time (min).
t_treat_	Treatment period is the time during which a constant temperature in the tumor (≥41 °C) is maintained (min).
Thermal Dose	
CEM43°CT_90_	Cumulative equivalent minutes at 43 °C when the measured temperature is T_90_ (min).
CEM43°CT_50_	Cumulative equivalent minutes at 43 °C when the measured temperature is T_50_ (min).
CEM43°CT_10_	Cumulative equivalent minutes at 43 °C when the measured temperature is T_10_ (min).
TRISE	T_50_ values above 37 °C multiplied by the duration of all heating sessions normalized to a duration of 450 min (°C) [36].
AUC	Actual time-temperature plots by computing the area under the curve (AUC) for T > 37 °C and T ≥ 39 °C (°C-min) [49].
HT sessions	
N_week_	Number of HT sessions per week.
N_total_	Total number of HT sessions during the treatment course.
Time interval	
t_int_	The time interval between HT and RT and/or CT.
Sequencing	The scheduling order of HT with RT and/or CT.

**Table 2 cancers-14-00625-t002:** Reference temperature metrics.

Temperature Metrics	Reference Value (°C)
T_min_	39
T_max_	44
T_avg_	Undefined
T_10_	Undefined
T_20_	Undefined
T_50_	≥41 *
T_80_	Undefined
T_90_	≥40 *

* According to ESHO guidelines for superficial HT [43].

**Table 3 cancers-14-00625-t003:** Reference heating duration parameters for HT.

Heating Duration Parameters	Reference Value (min)
t_pre_	undefined
t_treat_	60 ^1^

^1^ According to the Arrhenius plot [66].

**Table 4 cancers-14-00625-t004:** Reference thermal dose parameters for HT.

Thermal Dose Parameters	Reference Value
CEM43°CT_10_	Undefined (min)
CEM43°CT_50_	Undefined (min)
CEM43°CT_90_	Undefined (min)
TRISE	Undefined (°C)
AUC	Undefined (°C-min)

**Table 5 cancers-14-00625-t005:** Reference HT treatment session parameters. *N*: positive constant value.

Heating Session Parameter	Reference Value (*N*)
N_total_	Defined ^1^
N_week_	1–2 ^2^

^1^ Depending on RT and CT schedules; ^2^ Depending on cancer site.

**Table 6 cancers-14-00625-t006:** Reference t_int_ parameter for HT in combination with RT or CT.

Time Interval Parameter	Reference Value (min)
t_int_	0–240

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
