# Peer review of "Clinical Evidence for Thermometric Parameters to Guide Hyperthermia Treatment"

_cancers, 2022, doi:10.3390/cancers14030625_

Round 1

Reviewer 1 Report

This review summarizes the existing clinical evidence for the prognostic and predictive role of the most important thermometric parameters to guide combined treatment of RT and CT with HT. In conclusion, we call for standardization of thermometric parameters and stress the importance for their validation in future prospective clinical studies.

This is a well-written review paper, but a few comments need to be addressed.

  1. How these temperature parameters were measured, Tmin-Minimum temperature achieved in target volume (°C), Tmax-Maximum temperature achieved in target volume (°C), Tavg-Average temperature achieved in target volume (°C), T10-Temperature achieved in 10% of the target volume (°C), T20-Temperature achieved in 20% of the target volume (°C), T50-Temperature achieved in 50% of the target volume (°C), T80- Temperature achieved in 80% of the target volume (°C), T90-Temperature achieved in 90% of the target volume (°C)?
  2. The temperature parameters presented in this review are measurable in in vitro studies, but it is not pragmatic in clinical practices. Please address and discuss this issue.
  3. Hyperthermia therapy is widely used in clinical practices for tumor treatment, but the significantly used as thermal ablation for solid tumors, such as liver cancers, renal cancers, lung cancers and breast cancers etc. Please discuss thermal ablation for cancers in this review.
  4. How the hyperthermia was delivered to the tumor was not discussed. What instruments were used to produce the hyperthermia in the tumor?
  5. It is very hard to deliver homogeneous hyperthermia in tumors, which is dependent on what instruments are used to deliver the heat and how to achieve the precise, reliable and accurate measurement of temperatures in tumors. Please address and discuss this issue.

Reviewer 2 Report

This paper is a very valuable review of reports on the temperature factor in hyperthermia. Although a systematic review would be desirable, it may difficult at present due to the lack of uniformity in the definition of temperature factors, as reported.

Specific points

It would be better if the authors could provide a table or diagram of the recommended temperature factors and timing of heating.

”In a phase III study of the International Collaborative Hyperthermia Group, led by Vernon et al. [103], thermal dose was associated with complete response (CR) in patients treated for superficial recurrences of breast cancer [39]. Other studies failed to show a 3 relationship between outcome and thermal dose [99, 100, 111].”

It is problematic to make such above statements. This is because there are many positive results. Below important papers, such as those analysing data from below melanoma and cervical cancer RCTs, should be cited. .

Int J Hyperthermia. Jan-Feb 1996;12(1):3-20.

Int J Hyperthermia. 2018 Jun;34(4):461-468.

”Mitsumori et al. which failed to show improved treatment outcomes [83]. Different RT dose prescriptions and missing patient treatment data could underlie why this trial was negative and the authors stressed the need for internationally standardized treatment protocols for the combination of HT and RT.”

This Mitsumori's paper on a randomised controlled trial for lung cancer showed a significant improvement with regard to local control rates. This point should also be mentioned. It should also be mentioned that other below retrospective studies of regional hyperthermia in thoracic region have reported a significant association between intra-esophageal luminal temperature and treatment effect.

Int J Hyperthermia. 2011;27(1):20-6.

Int J Radiat Oncol Biol Phys. 2009 Jan 1;73(1):128-35.

”Temperature measurements in phantoms with MRT are accurate in comparison with thermistor probes [156] but clinical measurements are currently inaccurate in most pelvic and abdominal tumors [157].”

I would like to see a more detailed presentation of the problems for clinical measurements mentioned above.

The future of MRT and EPI should be shown in more detail.
